# In Vitro Regeneration, Acclimatization, Phytochemical Profiling, and Antioxidant Properties of Hong Hoen Sirirugsa (*Globba sirirugsae* Saensouk & P.Saensouk)

**DOI:** 10.3390/plants14223544

**Published:** 2025-11-20

**Authors:** Surapon Saensouk, Phiphat Sonthongphithak, Theeraphan Chumroenphat, Sukanya Nonthalee, Phannipha Phrommalee, Nooduan Muangsan, Toulaphone Keokene, Piyaporn Saensouk

**Affiliations:** 1Walai Rukhavej Botanical Research Institute, Mahasarakham University, Maha Sarakham 44150, Thailand; surapon.s@msu.ac.th; 2Diversity of Family Zingiberaceae and Vascular Plant for Its Applications Research Unit, Mahasarakham University, Maha Sarakham 44150, Thailand; phiphatmon@gmail.com; 3Cosmetic Science and Spa Program, Faculty of Thai Traditional and Alternative Medicine, Ubon Ratchathani Rajabhat University, Ubon Ratchathani 34000, Thailand; theeraphan.c@ubru.ac.th; 4Horticultural Research Institute, Trang Horticultural Research Center, Trang 92150, Thailand; sukanya_014@outlook.com; 5Department of Biology, Faculty of Science, Mahasarakham University, Maha Sarakham 44150, Thailand; phrommalee.p@gmail.com; 6School of Biology, Institute of Science, Suranaree University of Technology, Nakhon Ratchasima 30000, Thailand; nooduan@g.sut.ac.th; 7Department of Biology, Faculty of Natural Sciences, National University of Laos, Vientiane 7322, Laos; t.keokene@nuol.edu.la

**Keywords:** DPPH assay, HPLC analysis, micropropagation, ornamental plants, phenolics, β-pinene, volatile compounds, Zingiberaceae

## Abstract

*Globba sirirugsae* Saensouk & P.Saensouk, known in Thai as Hong Hoen Sirirugsa, is a rare Zingiberaceae species with considerable potential for ornamental horticulture and phytopharmaceutical development. Despite its promising attributes, comprehensive studies on its micropropagation, bioactivities, and phytochemical composition remain limited. This study investigated the efficiency of in vitro propagation using rhizome-derived plantlets cultured on Murashige and Skoog (MS) medium supplemented with various concentrations of BA, kinetin, and NAA. The highest shoot proliferation (5.67 shoots) was achieved with 4 mg/L BA and 0.5 mg/L NAA, while acclimatization in a soil–sand substrate (1:1) resulted in a 90% survival rate. Comparative analyses of wild and tissue-cultured plants revealed abundant phenolic and flavonoid contents, particularly in wild specimens, as determined by TPC and TFC assays. HPLC profiling confirmed the presence of bioactive compounds under both growth conditions. Ethanolic extracts exhibited strong antioxidant activities via 2,2-diphenyl-1-picrylhydrazyl (DPPH) and 2,2′-azino-bis(3-ethylbenzothiazoline-6-sulfonic acid) (ABTS) assays. GC-MS analysis identified 23 volatile compounds in wild plants and 51 in tissue-cultured plants, with α-pinene, β-pinene, caryophyllene, and α-bergamotene as dominant constituents. FTIR spectroscopy revealed distinct functional groups and fingerprint regions, serving as a rapid screening tool for phytochemical accumulation and biological activity. These findings provide a strategic foundation for the conservation and sustainable utilization of *Globba sirirugsae* as a multifunctional bioresource, with future applications in pharmaceutical innovation, product development, and ornamental landscaping.

## 1. Introduction

*Globba* L. is a genus in the Zingiberaceae family [1], traditionally used as medicinal herbs [2] and also popular as ornamental plants [3]. The *Globba* genus is primarily distributed in Southeast Asia, including Thailand, Myanmar, Laos, Malaysia, and Indonesia, making this region a center of its diversity [1,4].

*Globba sirirugsae* Saensouk & P.Saensouk (Figure 1), known as Hong Hoen Sirirugsa in Thai, is a perennial herb endemic to Thailand that was first discovered in Sakon Nakhon Province, Northeastern Thailand [5]. *Globba sirirugsae* has significant economic potential as an ornamental plant due to its attractive flowers; however, the limited natural population of this endemic species shows a declining trend as a result of ongoing environmental pressures and habitat disturbance, with unregulated commercial cultivation further threatening its survival. This species also undergoes seasonal dormancy, which limits conventional or natural propagation methods and impacts its consistent economic utilization, making effective propagation methods essential for conservation.

Previous studies on plant tissue culture in the *Globba* genus have demonstrated the potential for rapid propagation using exogenous plant growth regulators (PGRs) to optimize growth and multiplication. Tissue culture techniques have been successfully used to propagate various *Globba* species including *G. annamensis* [6], *G. brachyathera* [7], *G. globulifera* [8], *G. leucantha* var. *bicolor* [9], *G. marantina* [10,11,12], *G. schomburgkii* [11,13], *G. sherwoodiana* [14], and *G. williamsiana* [15], but research on the micropropagation and adaptation of this plant remains limited, highlighting a significant knowledge gap. Several species within the *Globba* genus have been documented for their traditional medicinal applications. For example, *G. marantina* is used as a remedy for conjunctivitis and in postnatal care [16], *G. multiflora* has analgesic and antipyretic properties [17], and *G. clarkei* is used to treat coughs [18]. Previous research also identified notable volatile compounds, phenolics, and flavonoids in *G. schomburgkii* [19,20], *G. ophioglossa* [20], *G. pendula* [21], *G. sessiliflora* [22], and *G. macrocarpa* [23], exhibiting antioxidant properties and biological activity.

Although *Globba sirirugsae* is horticulturally valuable and potentially useful as a medicinal herb, its in vitro propagation and phytochemical profile remain unexplored. This gap limits both conservation efforts and commercial development. To address this, we present the first protocol for in vitro propagation and phytochemical profiling of this endemic species, supporting its sustainable utilization and long-term preservation. This study examined the effects of plant growth regulators (PGRs) on shoot and root formation using solid MS medium, evaluated the influence of different substrates on plant acclimatization, and analyzed phytochemical profiles—including phenolic and flavonoid contents, antioxidant activity, volatile compositions, and functional group characteristics. Comparative analyses between wild plants and in vitro-cultured plants provide valuable insights into bioactive compound accumulation.

## 2. Results and Discussion

### 2.1. Effects of Plant Growth Regulators on the Micropropagation of Globba sirirugsae

#### 2.1.1. Effect of BA in Combination with NAA on Shoot and Root Production

To evaluate their regeneration potential, explants of *Globba sirirugsae* were cultured on solid Murashige and Skoog (MS) medium [24] supplemented with various concentrations of BA (1–5 mg/L) in combination with a constant concentration of NAA (0.1 mg/L). Growth parameters were recorded for all treatments after eight weeks of incubation. The addition of 6-benzylaminopurine (BA), either alone or in combination with 1-naphthaleneacetic acid (NAA), significantly enhanced plantlet regeneration compared with the control (MS medium without PGRs) (*p* < 0.05; Table 1, Figure 2). Application of BA alone at 3 mg/L resulted in the highest shoot induction (4.10 ± 0.57 shoots/explant). However, increasing BA concentrations above 3 mg/L caused a decline in plant multiplication, indicating that moderate exogenous BA levels effectively promoted cell division and organogenesis by enhancing endogenous auxin biosynthesis and maintaining auxin homeostasis in shoot meristems [25,26]. In contrast, excessive exogenous BA leads to overproduction of auxin, disrupting the balance between auxin biosynthesis and canalization/polar auxin transport [27]. Similarly, Parida et al. [12] reported that BA concentrations above 3 mg/L reduced the percentage of shoot proliferation in *Globba marantina*. while Yaowachai et al. [8] documented that explants of *Globba globulifera* cultured on BA alone at 3 and 5 mg/L exhibited the highest shoot formation.

The combination of BA with 0.1 mg/L NAA further promoted shoot regeneration, with the best response observed on medium containing 4 mg/L BA and 0.1 mg/L NAA, producing 5.10 ± 1.02 shoots/explant with a mean shoot length of 2.17 ± 0.21 cm (Figure 2). In terms of root development, the medium supplemented with 2 mg/L BA and 0.1 mg/L NAA exhibited the highest average root production (8.50 ± 2.20 roots/explant) with a root length of 1.53 ± 0.16 cm (Table 1). By contrast, the control medium yielded only 1.70 ± 0.26 shoots/explant with a mean shoot length of 1.47 ± 0.11 cm and 1.10 ± 0.62 roots/explant with an average root length of 0.36 ± 0.21 cm (Figure 2).

These results indicate that moderate exogenous BA concentrations are crucial for efficient in vitro regeneration of *Globba sirirugsae*, while the addition of NAA further enhanced both shoot and root formation. Cytokinins play a central role in plant tissue culture by promoting cell division, shoot regeneration, and developmental plasticity. Their application enhances in vitro propagation by activating networks of shoot-inducing genes and restoring totipotency in cultured explants [28,29]. Specifically, BA has been shown to promote shoot multiplication in several *Globba* species [6,7,8,10,11,12,13,14]. Auxins, such as NAA, support shoot and root morphogenesis, depending on their concentration and the surrounding hormonal context [25,27]. The combination of BA and NAA significantly enhanced shoot and root regeneration compared with BA alone. This synergistic interaction has also been observed in other *Globba* species, including *G. annamensis* [6], *G. brachyathera* [7], *G. marantina* [10,12], *G. schomburgkii* [13], and *G. sherwoodiana* [14].

#### 2.1.2. Effect of Kinetin in Combination with NAA on Shoot and Root Production

To evaluate *Globba sirirugsae* shoot and root production by exogenous kinetin and NAA, microshoots were cultured on solid medium containing either kinetin alone (1–5 mg/L) or kinetin in combination with NAA (0.1 mg/L) for 8 weeks. Compared to MS medium without PGRs, both kinetin alone and the combined application of kinetin and NAA significantly (*p* < 0.05) enhanced plant regeneration. (Table 2 and Figure 3). Solid MS medium supplemented with 4 mg/L kinetin exhibited the highest average shoot production of 3.90 ± 0.64 shoots/explant and mean shoot length of 2.89 ± 0.21 cm (Figure 3). These concentrations may stimulate endogenous auxin biosynthesis and maintain hormonal homeostasis, facilitating organized cell division. In contrast, the MS medium without PGRs produced only 1.90 ± 0.35 shoots/explant, although shoot length was slightly higher (2.95 ± 0.35 cm) (Figure 3), suggesting that kinetin primarily promotes shoot initiation rather than elongation.

The combination of 5 mg/L kinetin and 0.1 mg/L NAA demonstrated high root production, averaging 7.80 ± 1.24 roots/explant with mean root length 1.93 ± 0.13 cm (Table 2), highlighting the role of auxin–cytokinin synergy in root morphogenesis. NAA likely stabilizes cytokinin-induced responses and promotes root primordia formation.

Kinetin, a cytokinin-type phytohormone, is commonly applied in plant tissue culture to promote shoot and root induction. Kinetin alone promoted significantly higher shoot proliferation compared to the control medium. Increasing the concentration of kinetin in the medium impacted shoot production, with 5 mg/L kinetin decreasing the number of shoots per explant. This may result from excessive exogenous kinetin disrupting auxin canalization or polar transport pathways [27]. This result concurred with previous studies on *Globba marantina* [12], which reported limited effectiveness of high kinetin concentrations. By contrast, earlier studies on *G. annamensis* and *G. globulifera* found that high kinetin concentrations did not restrict shoot production and produced the highest shoot numbers [6,8]. Supplementation of kinetin with NAA markedly enhanced both shoot and root formation, although shoot length did not differ significantly. By contrast, kinetin alone promoted shoot formation over root induction. This synergistic effect aligned with Yaowachai et al. [8], who reported that the combination of kinetin and NAA at low concentrations significantly improved shoot length, root length, and root number in *G. globulifera* compared to kinetin alone at the same concentrations. These differences were attributed to species-specific responses or variations in culture conditions, including nutrient compositions, hormone types, and the physiological status of the explants.

#### 2.1.3. Influence of BA or Kinetin in Combinations with NAA on Shoot and Root Production

Microshoots were cultured on solid medium supplemented with BA or kinetin at varying concentrations, together with different levels of NAA, to investigate the interaction of hormone balance on shoot and root induction. The results are presented in Table 3 and Figure 4. The interaction of BA and NAA showed that medium containing 4 mg/L BA and 0.5 mg/L NAA produced the highest number of shoots, averaging 5.67 ± 0.46 shoots/explant with mean shoot length 3.98 ± 0.23 cm (Figure 4), while the highest shoot formation with kinetin in combination with NAA (4.00 ± 0.30 shoots, 3.41 ± 0.17 cm) was observed on medium supplemented with 5 mg/L kinetin and 0.5 mg/L NAA (Figure 4). By contrast, the control treatment (MS medium without PGRs) produced only 1.80 ± 0.20 shoots/explant and 20.70 ± 2.93 roots/explant (Figure 4). Among all the treatments, the highest root production (35.70 ± 0.70 roots/explant) was achieved in medium supplemented with 2 mg/L kinetin and 1 mg/L NAA, while the lowest (11.30 ± 1.82 roots/explant) was observed in 4 mg/L BA and 0.5 mg/L NAA (Table 3).

The combination of NAA with BA or kinetin promoted shoot and root induction, but increasing NAA to 1 mg/L reduced shoot production, particularly with BA, suggesting that superior efficacy of exogenous BA over kinetin in shoot development is attributed to its stronger cytokinin signaling, which activates key shoot-promoting genes [26,30]. Similarly, Muktawapai and Wongchaochant [14] reported that higher NAA levels (1–2 mg/L) combined with BA disrupted shoot production in *Globba sherwoodiana*. These findings contrasted with previous studies on *Globba* spp. [8,10,13], which reported that high NAA concentrations in combination with BA or kinetin did not inhibit shoot production. The variation in shoot and root formation among treatments reflects genotype-specific hormonal responses in plant tissue culture. Species and cultivars differ in their sensitivity to auxins and cytokinins due to inherent differences in hormone perception, signaling, and transport. Recent studies reported that variations in auxin canalization and cytokinin responsiveness among genotypes influence regeneration capacity and organ identity [30,31,32].

The developed protocol enables efficient shoot and root induction under controlled conditions, supporting the mass propagation of *Globba sirirugsae* for conservation and industrial purposes. Its scalability through bioreactor systems underscores the potential for phytopharmaceutical and cosmetic applications, while future studies should evaluate genetic stability and large-scale implementation.

### 2.2. Transplantation and Acclimatization

The effects of different substrates on the survival and growth parameters of *Globba sirirugsae* plantlets were evaluated during acclimatization (Table 4 and Figure 5). The mixed soil and sand substrate resulted in the highest survival rate of 90%, followed by sand alone with 80%, and soil alone with 70%. No significant differences in shoot numbers were observed among the substrates, with averages ranging from 2.20 to 2.50 shoots per explant. Similarly, the average number of leaves per explant did not differ significantly between treatments, ranging from 12.80 to 15.40 (Table 4). However, shoot length was significantly affected by substrate type. Plantlets grown in sand exhibited the highest shoot elongation, averaging 10.86 cm (Figure 5b), significantly longer than those grown in soil alone (8.05 cm) (Figure 5a) or in the 1:1 mixture of soil and sand (8.22 cm) (Figure 5c).

These results underscored the crucial importance of substrate selection during acclimatization to optimize water availability, aeration, and nutrient provision for the survival and subsequent growth of in vitro-derived plantlets. The superior survival observed in sand-mediated substrates was attributed to their improved aeration and drainage capacity, which reduced waterlogging and the risk of root rot, a major cause of plantlet mortality during acclimatization. Concurring with Saensouk et al. [13], *Globba schomburgkii* exhibited the highest acclimatization success on sand, with an 80% survival rate. Yaowachai et al. [8] reported that *G. globulifera* exhibited 100% survival on sand, soil, and a sand–soil mixture, while Parida et al. [12] found that *G. marantina* cultivated on a mixture of soil, cow dung, and sand exhibited a 70% acclimatization success rate with healthy morphology. Saensouk et al. [33] also reported that sand and a soil–sand mixture achieved 100% acclimatization success in *Curcuma larsenii* Maknoi & Jenjitikul. By contrast, soil alone hindered plantlet establishment due to its high water-holding capacity and low porosity, which restricted oxygen availability in the rhizosphere, thereby increasing physiological stress and susceptibility to disease.

Chlorophyll content, measured in SPAD units, also varied significantly across substrates (Table 4). The highest chlorophyll content was recorded in plantlets grown in sand alone (29.62 ± 1.26 SPAD units), followed by the soil–sand (1:1) mixture (24.70 ± 1.26 SPAD units), while the lowest chlorophyll content was observed in plantlets grown in soil alone (18.56 ± 2.11 SPAD units). The elevated chlorophyll levels observed in sand-grown plantlets indicated enhanced photosynthetic capacity and reduced physiological stress, likely due to improved aeration and drainage in the root zone. Similarly, Zucco et al. [34] reported that tomato plants cultivated in sandy soil amended with vermicompost exhibited improved growth performance, including increased plant height, leaf number, and SPAD value compared to other soil types.

Optimized substrates facilitated successful acclimatization of in vitro-derived plantlets, resulting in high survival and growth rates that support commercial-scale production by reducing transplant losses and enhancing vigor. Refining substrate formulation may further enhance early survival and metabolite accumulation. To ensure consistency in large-scale use, future studies should examine substrate effects on metabolite biosynthesis and stress tolerance, alongside clonal fidelity and genetic stability of acclimatized plants.

### 2.3. Evaluation of TPC, TFC, and Antioxidant Activity of Globba sirirugsae

The total phenolic content (TPC) and total flavonoid content (TFC) in different plant parts were evaluated under natural and tissue culture conditions (Table 5). Results showed that in wild plants, TPC ranged from 4.61 to 8.24 mg GAE/g DW, and TFC ranged from 1.61 to 3.10 mg RE/g DW, with the leaves segment exhibiting the highest values (8.24 mg GAE/g DW and 3.10 mg RE/g DW, respectively). By contrast, tissue-cultured plants contained lower levels of TPC and TFC, ranging from 3.76 to 4.45 mg GAE/g DW and 1.11 to 1.36 mg RE/g DW, respectively, with the highest contents observed in the roots segment (4.45 mg GAE/g DW and 1.36 mg RE/g DW, respectively). When comparing the leaf segments between the two growth conditions, the TPC value in wild plants was more than twice that found in tissue-cultured plants, with the TFC value in the roots and storage roots of wild plants higher than in tissue-cultured plants. These results suggested that phenolic and flavonoid compounds accumulated differentially across plant organs depending on the environmental growth conditions in natural habitats including light intensity, UV radiation, and temperature fluctuations, which are known to stimulate the biosynthesis of bioactive compounds as part of plant defense mechanisms against oxidative stress [35,36]. By contrast, tissue culture conditions provide a highly controlled environment with defined abiotic inputs including artificial nutrients and exogenous hormone regimes that may impose nutritional stress distinct from natural habitat [37]. These abiotic factors can trigger oxidative stress and subsequently upregulate key biosynthetic enzymes such as phenylalanine ammonia-lyase (PAL) and chalcone synthase (CHS), leading to enhanced accumulation of antioxidant compounds including phenolics and flavonoids [38]. Our findings concurred with previous reports. Yaowachai et al. [8] demonstrated that TPC and TFC in *Globba globulifera* varied significantly depending on plant organ and growth conditions. Similarly, Nonthalee et al. [39] observed that leaves of *Kaempferia grandifolia* and *K. siamensis* contained higher levels of these compounds in wild plants compared to tissue-cultured plants. Wild-derived *K. angustifolia* also exhibited higher TPC and TFC values than its in vitro counterparts [40].

The antioxidant activities of different plant parts were analyzed to assess their potential for future applications and to compare wild plants with tissue-cultured plants (Table 5). Results showed that wild plants possessed significantly higher antioxidant activity than tissue-cultured plants, as demonstrated by the DPPH and ABTS radical scavenging assays. The root and storage root segments of wild plants showed the strongest activity, with values of 5.23 mg TE/g DW in the DPPH assay and 88.85% of inhibition in the ABTS assay. By contrast, in tissue-cultured plants, the leaf segments exhibited the highest activity, with values of 4.10 mg TE/g DW in the DPPH assay and 61.31% of inhibition in the ABTS assay. Interestingly, across different explants, the ABTS assay consistently outperformed the DPPH scavenging assay in measuring antioxidant activity. Previous studies reported that the ABTS assay generally yielded higher antioxidant values than the DPPH assay, attributed to higher solubility of the ABTS•^+^ radical in aqueous and organic solvents. The dual electron and hydrogen-transfer mechanisms allowed interactions with a wide range of hydrophilic and lipophilic phenolics and flavonoids, while the stability of the ABTS•^+^ radical across physiological pH minimized spectral interference and reduced steric hindrance compared to the bulkier DPPH radical [41,42,43].

The higher antioxidant activity observed in naturally grown plants is closely associated with their elevated levels of phenolic and flavonoid compounds. The roots and storage roots function as storage organs and accumulate higher amounts of these secondary metabolites to protect against soil-borne pathogens and oxidative stress. By contrast, tissue-cultured plants grown under controlled in vitro conditions with defined abiotic inputs produced altered levels of bioactive compounds [37], resulting in reduced antioxidant capacity.

The Pearson correlation coefficients among TPC, TFC, and the antioxidant activities (DPPH and ABTS assays) are presented in Figure 6. A strongly positive correlation was observed between TPC and TFC (*r* = 0.82, *p* < 0.01). TFC also exhibited a strongly positive correlation with ABTS radical scavenging activity (*r* = 0.80, *p* < 0.01) and a moderate correlation with DPPH activity (*r* = 0.63, *p* < 0.01), suggesting that flavonoids contributed significantly to the antioxidant potential, particularly in the ABTS assay. The TPC showed a moderately positive correlation with both DPPH (*r* = 0.59, *p* < 0.01) and ABTS (*r* = 0.50, *p <* 0.05) assays, implying a notable but slightly weaker relationship compared to TFC. The DPPH and ABTS assays were moderately correlated (*r* = 0.69, *p* < 0.01), reflecting shared but assay-specific antioxidant mechanisms. While the analysis was based on a limited number of biological replicates, the observed correlations offer preliminary insights into potential associations between TPC/TFC value and antioxidant activity. These results concurred with previous studies that demonstrated a positive correlation between TPC/TFC and antioxidant activity [44,45,46] and provide a foundation for future validation with expanded datasets.

### 2.4. HPLC-Based Profiling of Phenolic Acids and Flavonoids in Globba sirirugsae

Qualitative and quantitative analyses of ethanolic extracts from various plant organs including leaves, pseudostems, rhizomes, roots, and storage roots of both wild and in vitro-cultured *Globba sirirugsae* were performed using HPLC (Table 6, Figure 7a). In wild plants, *p*-coumaric acid, cinnamic acid, and ferulic acid were detected across all examined parts. Concentrations of *p*-coumaric acid ranged from 138.90 to 141.33 μg/g DW, with the highest level observed in the leaves. Cinnamic acid levels ranged from 329.73 to 1447.67 μg/g DW, with the highest accumulation found in roots and storage roots. Ferulic acid concentrations ranged from 251.77 to 400.81 μg/g DW, with the highest content found in rhizomes. Caffeic acid was detected exclusively in rhizomes at 4.69 μg/g DW.

By contrast, in vitro-cultured plants exhibited a broader spectrum of phenolic acids, including caffeic acid, *p*-coumaric acid, cinnamic acid, ferulic acid, gallic acid, syringic acid, and vanillic acid. Vanillic acid was detected only in leaves and pseudostems, at concentrations of 442.12 and 230.19 μg/g DW, respectively. High levels of syringic acid were observed in roots (1418 μg/g DW) and leaves (917.61 μg/g DW).

Flavonoid profiling revealed the presence of kaempferol, quercetin, and rutin across all plant parts in both wild and tissue-cultured plants (Table 7, Figure 7b). In wild plants, kaempferol was most abundant in leaves (1755.62 μg/g DW), quercetin also peaked in leaves (434.26 µg/g DW), while rutin primarily accumulated in roots and storage roots (84.74 µg/g DW). In vitro-cultured plants exhibited a similar flavonoid profile, with quercetin and kaempferol showing particularly high levels in roots (802.55 and 552.44 μg/g DW, respectively).

This study presented the first comprehensive report of phenolic and flavonoid compounds in *Globba sirirugsae*. Our findings demonstrated that this species contained substantial amounts of bioactive compounds, notably cinnamic acid, ferulic acid, kaempferol, and quercetin, underscoring its potential as a valuable source of natural antioxidants. Among these, cinnamic acid and ferulic acid were detected in samples exhibiting strong radical scavenging activity, suggesting their direct contribution to the antioxidant potential. Cinnamic acid, a ubiquitous intermediate in the phenylpropanoid pathway commonly reported in medicinal plants [47], and ferulic acid, a hydroxycinnamic acid derivative detected in all plant parts, contributes to structural integrity by cross-linking with polysaccharides and lignin in cell walls [48]. These phenolic acids are known to donate hydrogen atoms and stabilize free radicals, which aligns with the elevated antioxidant activity recorded in our assays. Similarly, kaempferol and quercetin were abundant in treatments with high antioxidant capacity values, indicating their role in reducing DPPH and ABTS free radicals. Their hydroxylated flavonoid structures are capable of donating electrons, thereby enhancing redox potential under in vitro conditions.

Previous studies have demonstrated that the accumulation and biosynthesis of phenolic acid and flavonoid compounds are influenced by environmental factors, developmental stage, and genetic variability. For example, Wanyo et al. [49] reported high levels of chlorogenic acid and other phenolic acids in rhizomes of *Kaempferia parviflora*, while Saensouk et al. [50] observed variable phenolic and flavonoid contents in different plant parts of *Kaempferia larsenii* under diverse growth conditions. These findings aligned with the patterns observed in this study, highlighting the influence of growth conditions on secondary metabolite distribution.

The detection of phenolic and flavonoid compounds, confirmed by HPLC analysis and supported by notable antioxidant activity, underscores the medicinal relevance of *Globba sirirugsae* and its potential as a bioactive source for pharmaceutical and nutraceutical applications. The ability to produce these metabolites under controlled in vitro conditions offers a consistent and environmentally independent alternative to field-grown material. To further enhance metabolite yield and reliability, future research may focus on refining culture parameters—such as elicitor types and light regimes—and investigating the genetic basis of biosynthetic pathways to support targeted improvement and reproducibility.

### 2.5. Determination of Volatile Compounds in Globba sirirugsae by GC-MS Analysis

Gas chromatography-mass spectrometry (GC-MS) analysis of different plant parts of *Globba sirirugsae* revealed a detailed chemical profile of volatile organic compounds (Table 8), with 23 constituents identified in wild plants. The volatile compounds detected in the leaves of *G. sirirugsae* were primarily composed of sesquiterpene hydrocarbons (50.16%), followed by monoterpene hydrocarbons (41.86%) and oxygenated monoterpenes (2.51%). By contrast, the rhizomes and root-storage roots contained predominantly monoterpene hydrocarbons, accounting for 96.71% and 81.32% of the total volatiles, respectively. Sesquiterpene hydrocarbons were present in smaller proportions in these plant parts, contributing 0.85% in rhizomes and 18.68% in root-storage roots. Among the volatile compounds identified, monoterpene hydrocarbons constituted the dominant class in terms of relative abundance, particularly β-pinene (54.71%) in rhizomes and α-pinene (35.72%) in rhizomes. Caryophyllene (a sesquiterpene) was detected as a major constituent in roots and storage roots (18.68%). Other compounds, such as α-bergamotene (18.98%), δ-guaiene (6.05%), and copaene (5.56%), were exclusively detected in the leaves.

Compared to the wild plants, in vitro-cultured plants exhibited a broader volatile profile, with 51 identified compounds (Table 8). The volatile profile of the leaves from tissue-cultured plants was dominated by sesquiterpene hydrocarbons (69.19%), monoterpene hydrocarbons (14.23%), oxygenated sesquiterpenes (8.87%), and oxygenated monoterpenes (1.22%). Among the sesquiterpene hydrocarbons, caryophyllene and humulene were present in high amounts, accounting for 22.99% and 14.71%, respectively. In the pseudostem segment, the major constituents were sesquiterpene hydrocarbons (72.06%), monoterpenes (17.14%), and oxygenated monoterpenes (0.45%). The predominant components identified were β-pinene (11.50%) and cyperene (8.80%). Sesquiterpene hydrocarbons (53.73%) and monoterpene hydrocarbons (19.11%) were identified as the dominant groups in the roots of tissue-cultured plants, with cyperene (53.18%) and α-pinene (19.11%) being the main components.

These findings demonstrated that volatile oils from both wild and in vitro-cultured plants exhibited organ and condition-specific variation. In wild plants, the leaves were enriched in sesquiterpene hydrocarbons, whereas the rhizomes, roots and storage roots accumulated higher levels of monoterpene hydrocarbons. By contrast, tissue-cultured plants showed sesquiterpene hydrocarbons as the dominant volatiles across all organs. This diversity reflected the influence of multiple factors including genetic chemotypes, evolutionary adaptations, environmental and geographic conditions, developmental stages, organ-specific biosynthesis, harvest timing, seasonal variation, and extraction methods [51,52,53]. In addition, the nutrient composition of the culture medium may contribute to the observed differences. Changes in nutritional balance have been recognized to regulate secondary metabolite production and to significantly alter metabolite profiles. In particular, the high ammonium/nitrate ratio and salt content of MS medium are reported to induce nutritional stress and hyperhydricity, thereby influencing metabolite diversity in tissue-cultured plants [54].

Comparative studies on related Zingiberaceae species supported this variability. Rhizomes of *Alpinia pinnanensis* were rich in monoterpene hydrocarbons (33.02%) [55], whereas *Wurfbainia schmidtii* and *Zingiber atroporphyreus* showed monoterpenoids as the predominant constituents in both leaves and rhizomes [56]. By contrast, *Globba sessiliflora* contained mainly sesquiterpenoids (≈82.2%) with β-caryophyllene (20.1%), α-cadinol (14.7%), and selin-11-en-4α-ol (12%) as key compounds [22], while *G. marantina* was dominated by oxygenated monoterpenes (71.7%) alongside notable sesquiterpene hydrocarbons (22.11%) [57]. Similar patterns of sesquiterpenoid dominance were also reported in aerial parts of *Globba macrocarpa*, *Curcuma pierreana*, and *Zingiber pellitum* [23], as well as in rhizomes, leaves, and pseudostems of *G. schomburgkii* [19].

Volatile constituents, such as α-pinene and β-pinene, exhibit diverse pharmacological properties including anti-inflammatory, antimicrobial, and antitumor activities, and are extensively utilized in the flavor and fragrance industry [58]. Likewise, caryophyllene has shown remarkable potential for applications in pharmaceuticals, food and flavoring, and as an agricultural biopesticide [59], with α-bergamotene reported to contribute to flavor and fragrance products and crop protection [60]. Collectively, these findings highlight *Globba sirirugsae* as a valuable source of volatile oils with broad industrial and pharmaceutical potential.

### 2.6. Screening and Evaluation of Functional Groups in Different Plant Parts by FTIR Analysis

The FTIR technique was employed to identify the vibrational functional groups of organic compounds in *Globba sirirugsae* samples derived from the wild plants and in vitro-cultured plants (Table 9 and Figure 8). In both conditions, a broad absorption peak around 3330–3320 cm^−1^ was observed, corresponding to O–H stretching vibrations of alcohols and phenols [61]. Peaks observed around 2918–2929 cm^−1^ and near 2580 cm^−1^ were attributed to the asymmetric and symmetric C–H stretching vibrations of methylene (–CH_2_) groups associated with long aliphatic segments in cellulose and hemicellulose structures [62,63]. The region between 1730 and 1735 cm^−1^ indicated the presence of carbonyl (C=O) groups, suggesting esters or aldehydes [64]. The peaks at 1631–1642 cm^−1^ and 1516–1524 cm^−1^ were assigned to C=C stretching in aromatic groups commonly associated with phenolic compounds [62]. Bending vibrations of –CH_2_ and –CH_3_ groups were found at 1426–1396 cm^−1^ [63]. These peaks are commonly present in polysaccharide matrices. The band at 1360–1370 cm^−1^ was assigned to the symmetric bending vibrations of methyl (–CH_3_) groups [65] or O–H bending vibrations of alcohol and phenol groups [64]. The absorption peak at 1314–1320 cm^−1^ was primarily associated with O–H bending vibrations, potentially coupled with C–O stretching. These bands are typical in hydroxyl-rich structures such as phenolics, flavonoids, cellulose and other plant-derived polysaccharides [63,66]. C–O stretching vibrations were also observed at 1235–1244 cm^−1^ and 1032–1034 cm^−1^. The former corresponded to aryl ether linkages and phenolic C–O stretching, while the latter was associated with C–O stretching of primary alcohol groups. These functional groups are commonly found in cross-linked polyphenolic structures present in lignin [64,67].

## 3. Materials and Methods

### 3.1. Micropropagation of Globba sirirugsae

#### 3.1.1. Plant Collection and Explant Tissue Preparation

Plant Collection: *Globba sirirugsae* specimens were collected from Sakhon Nakhon Province, Thailand (July 2020) and used as initial explants for in vitro propagation.

Explant Tissue Preparation: Rhizomes from *Globba sirirugsae* were washed with running tap water for 30 min to remove impurities, disinfected with a disinfectant for 10 min, and then rinsed with clean water until all the impurities were removed. The cleaned rhizomes were sterilized in sodium hypochlorite (NaOCl) solution at various concentrations (20% (*v/v*) for 10 min, 10% (*v/v*) for 5 min). After sterilization, the rhizomes were washed several times with sterilized distilled water. For initial plantlet production, cleaned rhizomes were transferred onto Murashige and Skoog (MS) [24] medium at 25 ± 2 °C under 16 h photoperiod using fluorescent light at 27 μmol m^−2^ s^−1^. Medium preparation followed the method described by Saensouk et al. [50]. After 8 weeks of culture, the microshoots were subcultured onto solid MS medium supplemented with 2 mg/L BA (Sigma–Aldrich, Budapest, Hungary) and 0.1 mg/L NAA (SD Fine–Chem Limited, Mumbai, India) for 4 weeks. Finally, the plantlets obtained from the rapid growth medium were subcultured onto solid MS medium without PGRs for 4 weeks to maintain stable explants for experimental use.

#### 3.1.2. The Effect of PGRs on Plant Regeneration in MS Medium

The optimal conditions for shoot and root formation of *Globba sirirugsae* were evaluated on solid medium containing different PGRs. The explants, approximately 1 cm in length, from PGR-free medium-treated conditions were transferred onto solid MS medium containing BA (0, 1, 2, 3, 4, and 5 mg/L) in conjunction with NAA (0 and 0.1 mg/L) (Table 1). Explants were also cultured on solid MS medium supplemented with 6-furfurylaminopurine (kinetin, Sigma–Aldrich, Buchs, Switzerland) at concentrations of 0, 1, 2, 3, 4, and 5 mg/L, either alone or in combination with NAA at 0 and 0.1 mg/L (data presented in Table 10). A range of cytokinin and auxin concentrations, from low to high, was also evaluated to determine the optimal conditions for plant formation. Various combinations of BA or kinetin with NAA at different concentrations were tested to assess their effects. The explants from PGR-free medium-treated conditions were inoculated on solid MS medium containing BA or kinetin at concentrations ranging from 0 to 5 mg/L, in combination with NAA at 0, 0.5, and 1 mg/L. The medium formulation data are shown in Table 10. Each treatment was replicated 10 times to present mean values of the dependent variables such as number of shoots per explant, number of roots per explant, shoot length, and root length.

### 3.2. Transplantation and Acclimation

The in vitro-cultured plants of *Globba sirirugsae* were transplanted under different planting materials to observe and establish their adaptation. Firstly, plantlets from in vitro culture on solid MS medium (PGR-free medium) were cultured for 8 weeks and then acclimatized at room temperature for 2 weeks before transfer to the planting materials. Strong and healthy pre-acclimation-plantlets (5–6 cm long) were washed with tap water. After cleaning, the plantlets were cultivated in plastic pots containing different planting materials: soil, sand, and a 1:1 (*w*/*w*) mixture of soil and sand. Each planting material was tested with 10 replications. The plantlets were kept in a greenhouse at Mahasarakham University (Department of Biology, Faculty of Science) for an 8-week acclimatization period. During the experimental period, the acclimatization conditions were closely monitored. The plantlets were frequently watered with tap water to prevent dehydration and maintain optimal moisture levels. After transplanting the plantlets into growth substances materials, the growth parameters were recorded to assess adaptation to condition regeneration, including survival rate (percentage survival of acclimated plants), average shoot production, leaf production, and shoot height. The chlorophyll content in the leaves was estimated using a chlorophyll meter (SPAD-502Plus, Konica Minolta Inc., Osaka, Japan) to assess the effect of different growing materials on chlorophyll content and plant development, following the method of Wicharuck et al. [68]. To measure the chlorophyll content in the leaves, the SPAD-502Plus was calibrated before recording the transmittance. Leaf samples from each growth condition were assessed with 10 replications and reported as soil–plant analysis development (SPAD) values. Measurements were conducted at the central portion of the leaf blade, midway between the midrib and the leaf margin, to ensure consistency and accuracy.

### 3.3. Determination of Phytochemical Compounds and Antioxidant Activity

#### 3.3.1. Plant Material and Preparation of Plant Extraction

Plant Material: Tissue-cultured plantlets of *Globba sirirugsae* were collected after culturing for 8 weeks on solid MS medium containing 2 mg/L BA and 0.1 mg/L NAA. Various plant parts of tissue-cultured plants, including leaves, pseudostems, and roots, were separated for extraction. Wild-grown plants were collected from Sakon Nakhon Province, Thailand, in July 2021. The specimens were identified and authenticated by Associate Professor Dr. Surapon Saensouk at the Walai Rukhavej Botanical Research Institute (WRBRI), Mahasarakham University, Thailand. Voucher specimens were deposited in the Vascular Plant Herbarium, Mahasarakham University (VMSU) in Thailand. Several plant parts, including leaves, pseudostems, roots and storage roots, and rhizomes were also prepared for future extraction.

Plant Preparation and Extraction: Each plant part from both growth conditions (wild plants and tissue-cultured plants) was washed several times with tap water, followed by rinsing with distilled water. The cleaned materials were then freeze-dried for 48 h to remove moisture. The dried samples were ground into a fine powder and stored at −20 °C until use. The powders obtained from the different plant parts were extracted using the maceration method, following Chumroenphat et al. [69] with slight modifications. Briefly, 0.3 g of plant powder was mixed with 20 mL of 80% (*v/v*) ethanol and incubated at 37 °C for 15 h with shaking at 150 rpm. After incubation, the ethanolic extract was filtered through filter paper (Whatman^®^ No. 1). The filtrate was stored at −20 °C for use in subsequent experiments.

#### 3.3.2. Total Phenolic Content (TPC)

The Folin–Ciocalteu assay was used to determine the total phenolic content in the plant extracts, following the protocol established by Saensouk et al. [50]. Firstly, 100 µL of Folin–Ciocalteu reagent (20% *v/v*, Sigma–Aldrich, Munich, Germany) was mixed with plant extract (20 µL) in a 96-well plate. Next, the mixture was incubated with shaking at room temperature for 5 min. Then, 75 µL of 10% (*w/v*) sodium carbonate (Na_2_CO_3_) solution was added and further incubated in the dark at room temperature for 2 h. Finally, the absorbance of the mixture solution was recorded by a UV–Vis microplate reader (Variokan™ LUX, Thermo Fisher Scientific Inc., Marsiling, Singapore) at 750 nm. A calibration curve was prepared using gallic acid as the standard (Merck KGaA, Beijing, China) to calculate the total phenolic content, expressed as milligrams of gallic acid equivalent per gram of dry weight (mg GAE/g DW).

#### 3.3.3. Total Flavonoid Content (TFC)

The aluminum chloride colorimetric method was used to evaluate the total flavonoid content in the different plant parts following Saensouk et al. [50] In brief, 10 µL of a 5% (*w/v*) sodium nitrite (NaNO_2_) solution was mixed with deionized water (100 µL) in a 96-well plate containing ethanolic extract (25 µL). The mixture was incubated with shaking for 5 min and then 15 µL of 10% (*w/v*) aluminum chloride hexahydrate (AlCl_3_·6H_2_O) solution was added, with the incubation continued for 6 min in the dark. Next, 50 µL of 1 M sodium hydroxide (NaOH) and 50 µL of deionized water were added to complete the reaction. The absorbance of the mixture solution was measured at a wavelength of 510 nm using a UV–Vis microplate reader. To calculate the TFC, a calibration curve of the rutin standard (Sigma–Aldrich, Beijing, China) solution was used, with the results expressed as milligrams of rutin equivalent per gram of dry weight (mg RE/g DW).

#### 3.3.4. DPPH Radical Scavenging Assay

The 1,1-diphenyl-2-picrylhydrazyl (DPPH) radical was used to evaluate the antioxidant properties of the ethanolic extracts. This assay was performed based on the method of Rivero-Perez et al. [70] with slight modifications. Firstly, 20 µL of ethanolic extract was mixed with 150 µL of 0.15 M DPPH radical (Alfa Aesar, Thermo Fisher Scientific, Waltham, MA, USA) solution in a 96-well plate. The mixture was then homogenized and incubated in the dark at room temperature for 30 min. Finally, the mixture was recorded for spectra at a wavelength of 517 nm using a UV–Vis microplate reader. A Trolox standard (Sigma–Aldrich, Munich, Germany) solution was used to calculate the DPPH radical scavenging activity. The calibration curve of Trolox was expressed as milligrams of Trolox equivalent per gram of dry weight of sample (mg TE/g DW). The percentage inhibition of DPPH free radicals was calculated by the following equation:(1)$$\%\;\mathrm{inhibition}\;=\;[(A_{B}\;-\;A_{E})/A_{B}]\;\times \;100$$
where A_B_ is the absorbance of the blank solution (80% ethanol), and A_E_ is the absorbance of the ethanolic extract after reaction with the DPPH radical solution.

#### 3.3.5. ABTS Radical Scavenging Capacity

The 2,2′-azino-bis(3-ethylbenzthiazoline-6-sulfonic acid) (ABTS) assay was conducted as described by Payet et al. [71] with some modifications. First, an ABTS radical cation solution was prepared by mixing a 7 mM ABTS (Sigma–Aldrich, Munich, Germany) solution with a 2.45 mM potassium persulfate (K_2_S_2_O_8_, QRec Chemical Co., Ltd., Auckland, New Zealand) solution in a 1:2 ratio, followed by incubation in the dark at 25 °C for 12–16 h. The solution was prepared in 50% (*v/v*) ethanol. Then, the ABTS radical cation solution was diluted with 50% (*v/v*) ethanol to achieve an absorbance of 0.700 ± 0.02 at a 734 nm wavelength and equilibrated at 30 °C. To determine the total antioxidant capacity, 180 µL of the diluted ABTS radical cation solution was added to 20 µL of the ethanolic extract in a 96-well plate. The mixture was mixed and incubated for 5 min and the absorbance was measured at 734 nm using a UV–Vis microplate reader. The percentage inhibition of the ABTS radical was calculated as follows:(2)$$\%\;\mathrm{inhibition}\;=\;[(T_{B}\;-\;T_{E})/T_{B}]\;\times \;100$$
where T_B_ is the absorbance of the blank solution (80% ethanol), and T_E_ is the absorbance of the ethanolic extract after reaction with the ABTS radical cation solution.

### 3.4. Identification and Quantitative Analysis of Phenolic Acids and Flavonoid Components by the HPLC Assay

Plant Sample Preparation: Both wild-grown and in vitro-cultured plants were prepared to compare their contained phytochemicals in different plant parts. The leaves, pseudostems, roots and storage roots, and rhizomes of wild-grown plants were sampled, while the leaves, pseudostems, and roots of in vitro-cultured plants were sampled. All the samples were prepared using the method described in Section 3.3.1. The plant extracts were prepared following the method outlined by Chumroenphat et al. [72] with slight modifications. Briefly, 0.3 g of dried powder from the different plant parts was macerated with 20 mL of 1% (*v/v*) hydrochloric acid (prepared in methanol). The mixtures were then incubated at 37 °C, with shaking at 150 rpm for 15 h. After incubation, the mixtures were filtered using Whatman^®^ No. 1 filter paper, followed by filtration through a 0.22 µm nylon membrane before HPLC analysis.

Chromatographic Conditions: The analysis was performed on a Shimadzu LC–20A series system (Shimadzu Corp., Tokyo, Japan) comprising a Shimadzu LC–20AC pump and an SPD-M20A diode array detector. Chromatographic separation was achieved on a C–18 column (4.6 mm × 250 mm, 5 µm; LUNA^®^, Phenomenex Inc., Torrance, CA, USA) protected by a guard column. The mobile phase composition was programmed at gradient elution conditions comprising 0.1% (*v/v*) acetic acid (solvent A) and acetonitrile (solvent B), with a flow rate of 0.8 mL/min. The gradient elution system followed the method described by Siriamornpun and Kaewseejan [73], with the chromatographic conditions shown in Table 11.

The phenolic acid and flavonoid contents in the various plant parts were identified and quantified by comparing their retention times (RTs) and UV spectra with authentic standards. The quantification was conducted using the external standard method.

### 3.5. GC-MS Analysis of Volatile Components in Globba sirirugsae

Plant Sample Preparation: The samples were collected from plants grown under different conditions, including wild plants and plantlets cultured in vitro on solid MS medium supplemented with 2 mg/L BA and 0.1 mg/L NAA for 8 weeks. All the samples were extracted following Nonthalee et al. [39]. The vials were sealed with an aluminum-rubber septum (Supelco, Bellefonte, PA, USA).

GC-MS Chromatographic Conditions: The volatile compounds under different plant growing conditions were identified following the method of Nonthalee et al. [39] using a GC-MS series QP-2010 (Shimadzu, Tokyo, Japan). Chromatographic separation was achieved on a fused silica capillary column Rtx-5Ms (5% diphenyl 95% dimethyl polysiloxane, 30 mm length, 0.25 mm internal diameter, 0.25 µm film thickness; Restek, Bellefonte, PA, USA). The conditions were set as follows: carrier gas, helium (1.0 mL/min), with a split ratio of 10:1; injector temperature, 280 °C. The temperature program was set to 70–280 °C at a rate of 5 °C/min, followed by an isothermal hold at 280 °C for 10 min. The temperature of the transfer line theater was set at 280 °C. The ionization energy was set at 70 eV, and electron ionization mass spectra were acquired over a mass range of 50–550 amu. Data acquisition and peak integration were performed using GC-MS Solution software version 2.53 (Shimadzu, Tokyo, Japan), with further data processing conducted using Microsoft Excel 365 (Microsoft, Washington, DC, USA). Individual components were identified based on their mass spectra using the National Institute of Standards and Technology (NIST 11) libraries, NIST Chemistry Webbook [74], and Adam’s book [75].

### 3.6. Screening and Evaluation of Molecular Functional Groups of Phytochemicals in Globba sirirugsae by the FTIR Technique

FTIR spectroscopy was used for preliminary screening of the bioactive compounds, and identification of the functional groups in various plant parts, to determine the vibrations of molecules excited by infrared radiation within the fingerprint region specific to each compound. This assay rapidly analyzed the functional groups in medicinal plants and simplified sample preparation for evaluation. To assess the functional groups of each plant part, dried powder from Section 3.3.1 was measured using a UATR accessory (PerkinElmer, Waltham, MA, USA) on a Diamond/KRS–crystal composite for spectrum recording. The infrared spectrum data were collected from 32 scans at a resolution of 4 cm^−1^. The wavenumber range was set from 4000 to 400 cm^−1^, with background subtraction performed automatically by the software, following the method described by Chumroenphat et al. [72].

### 3.7. Statistical Analysis

Micropropagation: The tissue culture study was conducted using a completely randomized design (CRD). A Shapiro–Wilk test was performed to evaluate the normality of the data, and homogeneity of variances was assessed using Levene’s test. All outcomes of dependent variables were reported as mean ± standard error (SE), with each treatment replicated ten times. Data were analyzed using one-way analysis of variance (ANOVA) at a significance level of *p* < 0.05, and mean differences were determined by Duncan’s multiple range test (DMRT) using SPSS statistical software (IBM Corp., New York, NY, USA; version 29).

Phytochemical Profiling and Antioxidant Capacity Analysis: All the results were reported as mean ± standard error (SE), with each treatment replicated three times. Data were analyzed using one-way ANOVA, followed by DMRT for mean comparison, with the significance level set at *p* < 0.05 using SPSS statistical software. The Pearson correlation coefficient (*r*) was also calculated to assess the strength and direction of the relationships between variables, including TPC, TFC, DPPH, and ABTS values.

## 4. Conclusions

This study demonstrated the biological and phytochemical richness of *Globba sirirugsae*, underscoring its value as a regionally significant yet underexplored species. The successful establishment of micropropagation protocols and comparative phytochemical profiling provided a scientific basis for both its conservation and sustainable utilization. Our findings highlighted the potential of this species for applications in herbal medicine, cosmetics, and ornamental horticulture, while emphasizing the influence of growth conditions on the expression of secondary metabolites. Future research should further optimize cultivation strategies to enhance the yield of bioactive compounds for high-value product development.

## Figures and Tables

**Figure 1 plants-14-03544-f001:**
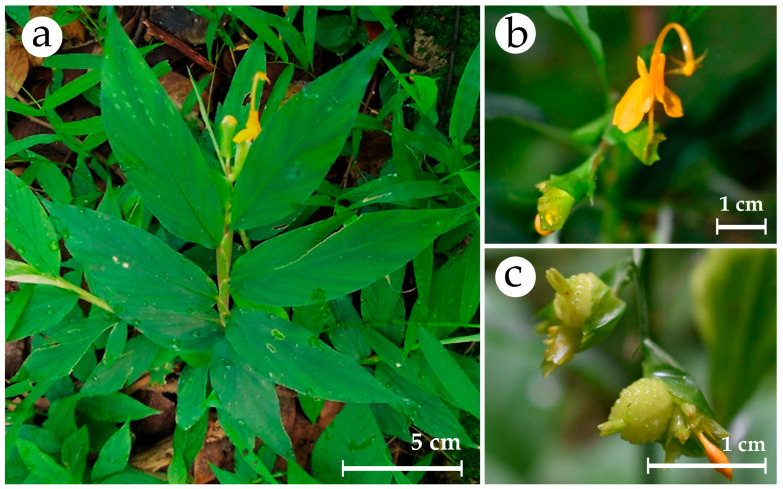
*Globba sirirugsae* Saensouk & P.Saensouk. (**a**) plant in habitat; (**b**) flower; (**c**) fruits. Scale bar: 5 cm (**a**); 1 cm (**b**,**c**).

**Figure 2 plants-14-03544-f002:**
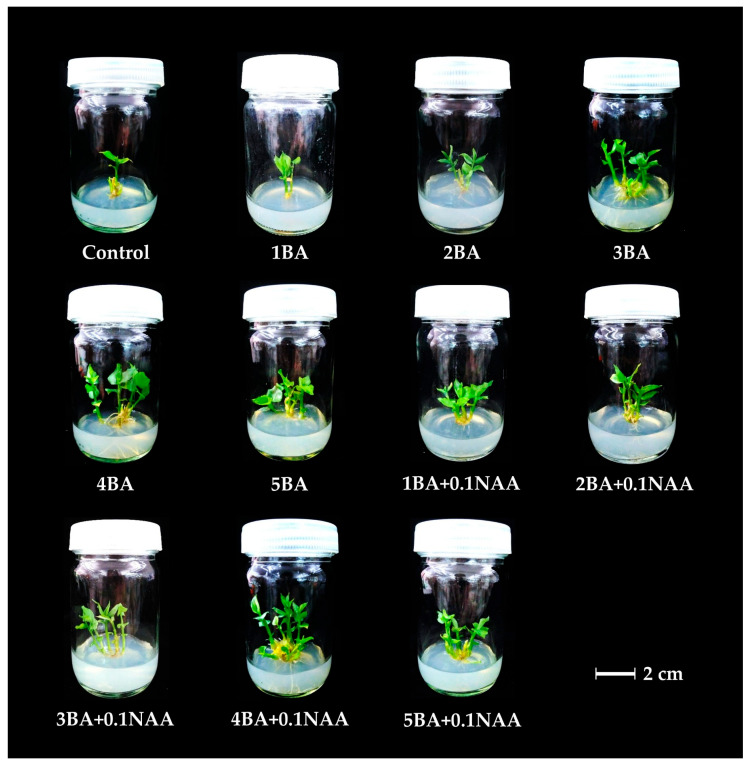
Effect of BA and NAA on in vitro plantlet regeneration of *Globba sirirugsae*; Scale bar = 2 cm.

**Figure 3 plants-14-03544-f003:**
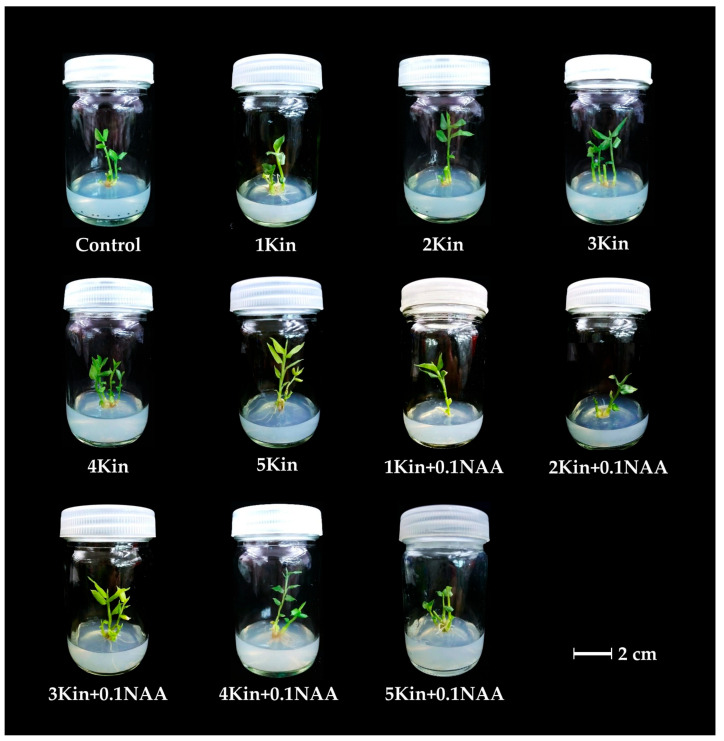
Effect of kinetin and NAA on in vitro plantlet regeneration of *Globba sirirugsae*; Scale bar = 2 cm.

**Figure 4 plants-14-03544-f004:**
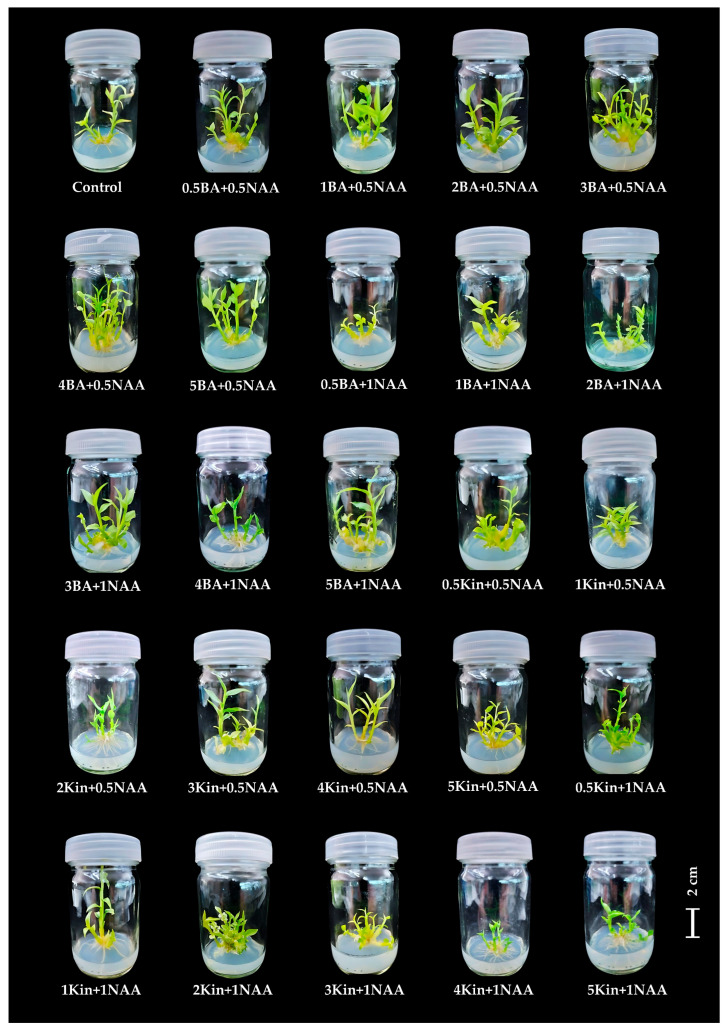
Effect of BA or kinetin in combination with NAA at various concentrations on in vitro regeneration of *Globba sirirugsae*; Scale bar = 2 cm.

**Figure 5 plants-14-03544-f005:**
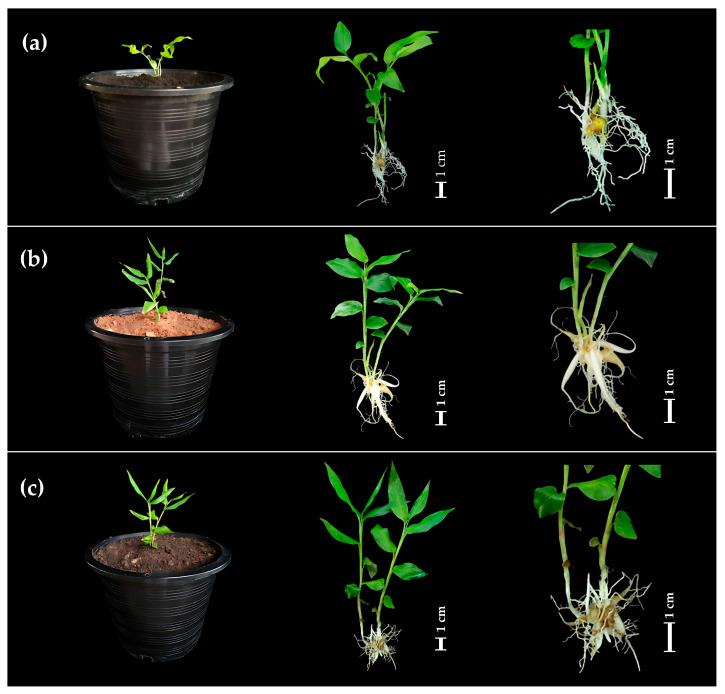
Effect of planting substrates on the development of in vitro-cultured *Globba sirirugsae* plantlets: (**a**) soil; (**b**) sand; (**c**) soil–sand (1:1). Scale bar = 1 cm.

**Figure 6 plants-14-03544-f006:**
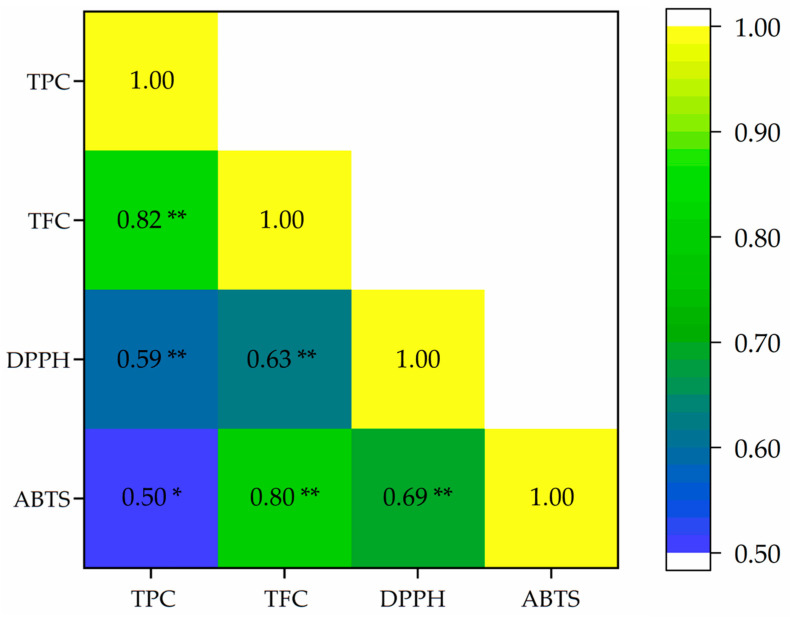
Pearson correlation heatmap (*n* = 7) among TPC, TFC, DPPH, and ABTS in *Globba sirirugsae*. Color intensity indicates correlation strength (*r* = 0.50–1.00). * *p* < 0.05, ** *p <* 0.01.

**Figure 7 plants-14-03544-f007:**
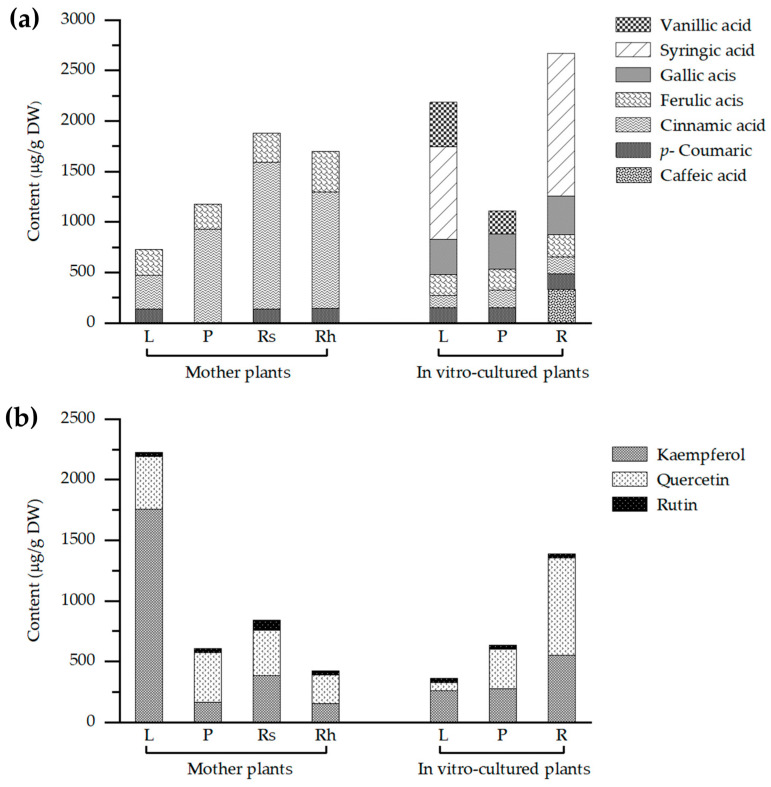
Quantitative HPLC analysis of (**a**) phenolic acids and (**b**) flavonoid compounds in different plant parts of *Globba sirirugsae* from mother plants and in vitro-cultured plants. L = leaves; P = pseudostems; Rh = rhizomes; R = roots; Rs = roots and storage roots.

**Figure 8 plants-14-03544-f008:**
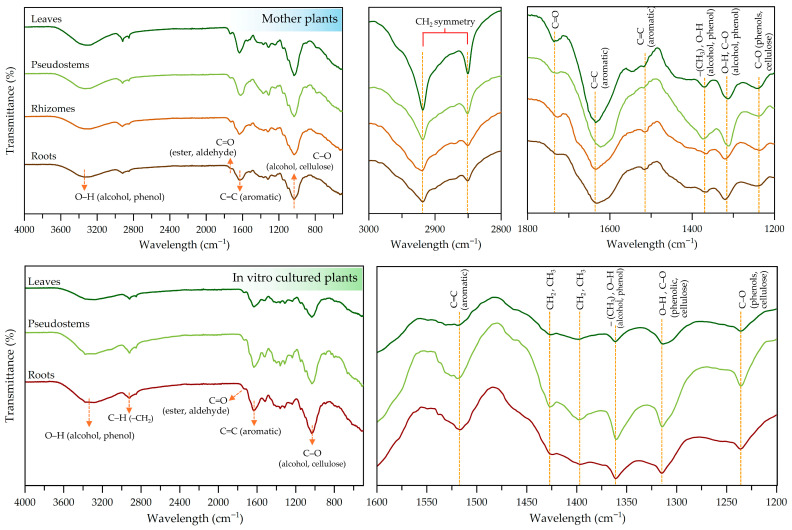
FTIR analysis of various plant parts of *Globba sirirugsae* grown under different conditions.

**Table 1 plants-14-03544-t001:** Effects of BA and NAA on the regeneration of *Globba sirirugsae* cultured on solid medium for 8 weeks.

BA (mg/L)	NAA (mg/L)	Average No. of Shoots/Explant	Average Shoot Length (cm)	Average No. of Roots/Explant	Average Root Length (cm)
0	0	1.70 ± 0.26 b	1.47 ± 0.11 b	1.10 ± 0.62 c	0.36 ± 0.21 c
1	0	1.80 ± 0.33 b	1.83 ± 0.20 ab	1.51 ± 0.12 c	0.29 ± 0.03 c
2	0	3.30 ± 0.42 ab	1.97 ± 0.11 ab	1.40 ± 0.37 c	0.25 ± 0.07 c
3	0	4.10 ± 0.57 a	1.86 ± 0.15 ab	2.10 ± 0.99 bc	0.30 ± 0.13 c
4	0	3.80 ± 0.70 ab	1.68 ± 0.21 ab	1.10 ± 0.60 c	0.11 ± 0.05 c
5	0	3.10 ± 0.55 ab	1.49 ± 0.22 b	1.40 ± 0.90 c	0.19 ± 0.80 c
1	0.1	3.00 ± 0.52 ab	1.91 ± 0.16 ab	7.30 ± 1.02 a	1.10 ± 0.11 ab
2	0.1	3.33 ± 0.56 ab	2.03 ± 0.15 ab	8.50 ± 2.20 a	1.53 ± 0.16 a
3	0.1	3.80 ± 0.98 ab	1.98 ± 0.16 ab	5.00 ± 1.15 ab	0.96 ± 0.21 b
4	0.1	5.10 ± 1.02 a	2.17 ± 0.21 a	7.80 ± 1.53 a	1.11 ± 0.20 ab
5	0.1	3.60 ± 0.86 ab	1.69 ± 0.19 ab	6.50 ± 1.61 a	0.86 ± 0.24 b

Values are mean ± SE from ten biological replicates. Means with the same letters in each column are not significantly different at *p* ≤ 0.05 according to DMRT.

**Table 2 plants-14-03544-t002:** Effects of kinetin and NAA on the regeneration of *Globba sirirugsae* cultured on solid medium for 8 weeks.

Kinetin (mg/L)	NAA (mg/L)	Average No. of Shoots/Explant	Average Shoot Length (cm)	Average No. of Roots/Explant	Average Root Length (cm)
0	0	1.90 ± 0.35 bc	2.95 ± 0.35 a	0.80 ± 0.53 d	0.21 ± 0.17 d
1	0	2.00 ± 0.00 bc	2.93 ± 0.34 a	1.90 ± 0.43 cd	0.70 ± 0.10 cd
2	0	1.60 ± 0.27 bc	3.42 ± 0.35 a	0.80 ± 0.36 d	0.33 ± 0.14 cd
3	0	2.30 ± 0.42 bc	3.03 ± 0.33 a	0.30 ± 0.21 d	0.20 ± 0.13 d
4	0	3.90 ± 0.64 a	2.89 ± 0.21 a	2.10 ± 0.64 bcd	0.78 ± 0.15 cd
5	0	2.10 ± 0.50 bc	3.67 ± 0.29 a	2.30 ± 0.73 bcd	0.81 ± 0.13 c
1	0.1	1.20 ± 0.13 c	3.15 ± 0.37 a	3.30 ± 0.82 bc	1.41 ± 0.27 b
2	0.1	1.70 ± 0.21 bc	2.73 ± 0.34 a	3.80 ± 0.77 bc	1.77 ± 0.17 ab
3	0.1	1.70 ± 0.34 bc	3.01 ± 0.31 a	4.50 ± 1.14 b	2.09 ± 0.33 a
4	0.1	1.80 ± 0.42 bc	3.46 ± 0.31 a	4.40 ± 1.05 b	1.95 ± 0.23 ab
5	0.1	2.90 ± 0.64 ab	2.85 ± 0.23 a	7.80 ± 1.24 a	1.93 ± 0.13 ab

Values are mean ± SE from ten biological replicates. Means with the same letters in each column are not significantly different at *p* ≤ 0.05 according to DMRT.

**Table 3 plants-14-03544-t003:** Effect of BA or kinetin in combination with NAA at various concentrations on in vitro regeneration of *Globba sirirugsae* cultured on solid medium for 8 weeks.

BA (mg/L)	Kinetin (mg/L)	NAA (mg/L)	Average No. of Shoots/Explant	Average Shoot Length (cm)	Average No. of Roots/Explant	Average Root Length (cm)
0	0	0	1.80 ± 0.20 f	3.93 ± 0.28 ab	20.70 ± 2.93 c	1.90 ± 0.15 ab
0.5	0	0.5	3.67 ± 0.24 c	3.45 ± 0.17 ab	20.80 ± 0.94 c	2.03 ± 0.05 a
1	0	0.5	4.40 ± 0.50 ab	4.20 ± 0.30 a	16.30 ± 1.84 e	2.04 ± 0.04 a
2	0	0.5	4.10 ± 0.41 ab	4.59 ± 0.37 a	14.30 ± 1.94 e	2.06 ± 0.13 a
3	0	0.5	4.89 ± 0.51 a	4.38 ± 0.45 a	12.00 ± 1.90 f	1.68 ± 0.10 b
4	0	0.5	5.67 ± 0.46 a	3.98 ± 0.23 ab	11.30 ± 1.82 f	1.90 ± 0.17 ab
5	0	0.5	4.12 ± 0.42 ab	4.62 ± 0.31 a	11.70 ± 1.58 f	1.94 ± 0.12 ab
0.5	0	1	2.80 ± 0.25 de	3.08 ± 0.27 b	20.80 ± 0.94 c	1.76 ± 0.13 ab
1	0	1	3.10 ± 0.18 c	3.31 ± 0.17 b	13.90 ± 1.02 f	2.22 ± 0.22 a
2	0	1	4.20 ± 0.29 ab	3.18 ± 0.25 b	16.80 ± 2.22 e	1.12 ± 0.14 cd
3	0	1	4.10 ± 0.38 ab	4.31 ± 0.27 a	19.80 ± 2.78 c	1.58 ± 0.21 b
4	0	1	4.22 ± 0.36 ab	4.05 ± 0.33 a	16.89 ± 2.47 e	1.21 ± 0.15 cd
5	0	1	4.00 ± 0.30 ab	4.17 ± 0.26 a	22.40 ± 1.87 b	1.53 ± 0.12 b
0	0.5	0.5	2.89 ±0.31 de	3.10 ± 0.30 b	19.78 ± 1.26 c	1.21 ± 0.08 cd
0	1	0.5	3.30 ± 0.33 c	3.42 ± 0.32 ab	21.10 ± 3.06 bc	0.99 ± 0.14 d
0	2	0.5	2.44 ± 0.34 de	3.87 ± 0.49 a	20.89 ± 2.68 c	1.67 ± 0.21 b
0	3	0.5	2.70 ± 0.21 de	3.81 ± 0.41 a	17.20 ± 2.32 d	1.36 ± 0.22 c
0	4	0.5	2.80 ± 0.25 de	3.68 ± 0.25 a	20.80 ± 2.34 c	1.34 ± 0.19 c
0	5	0.5	4.00 ± 0.30 ab	3.41 ± 0.17 ab	23.30 ± 1.74 b	1.48 ± 0.12 c
0	0.5	1	3.20 ± 0.20 c	2.98 ± 0.22 c	21.90 ± 1.15 bc	1.48 ± 0.12 c
0	1	1	2.70 ± 0.40 de	4.18 ± 0.59 a	18.10 ± 3.64 d	0.84 ± 0.19 d
0	2	1	3.90 ± 0.35 ab	4.32 ± 0.25 a	35.70 ± 2.67 a	1.19 ± 0.14 cd
0	3	1	3.22 ± 0.36 c	3.70 ± 0.47 a	22.00 ± 4.95 b	1.23 ± 0.34 cd
0	4	1	3.56 ± 0.18 c	2.79 ± 0.29 c	15.44 ± 3.04 e	0.65 ± 0.13 d
0	5	1	3.67 ± 0.24 c	2.97 ± 0.23 c	18.00 ± 2.07 d	0.92 ± 0.13 d

Values are mean ± SE from ten biological replicates. Means with the same letters in each column are not significantly different at *p* ≤ 0.05 according to DMRT.

**Table 4 plants-14-03544-t004:** Effect of different planting substrates on survival rate and growth of *Globba sirirugsae* after 8 weeks of cultivation in the greenhouse.

Plant Material	Percentage of Surviving Plantlets (%)	Average No. of Shoots/Explant	Average No. of Leaves/Explant	Average Shoot Length (cm)	Chlorophyll Content (SPAD Unit)
Soil	70	2.30 ± 0.30 a	12.80 ± 1.00 a	8.05 ± 0.88 b	18.56 ± 2.11 c
Sand	80	2.20 ± 0.25 a	14.50 ± 1.12 a	10.86 ± 0.79 a	29.62 ± 1.26 a
Soil–sand	90	2.50 ± 0.17 a	15.40 ± 0.96 a	8.22 ± 1.01 b	24.70 ± 1.26 b

Values are mean ± SE from three technical replicates. Means with the same letters in each column are not significantly different at *p* ≤ 0.05 according to DMRT.

**Table 5 plants-14-03544-t005:** TPC, TFC, DPPH and ABTS values of wild plants and in vitro-cultured plants of *Globba sirirugsae*.

Condition	Part of Plant	TPC (mg GAE/g DW)	TFC (mg RE/g DW)	DPPH (mg TE/g DW)	ABTS (% Inhibition)
wild plants	Leaves	8.24 ± 0.07 a	3.10 ± 0.22 a	3.06 ± 0.03 c	70.35 ± 0.58 c
	Pseudostems	4.61 ± 0.06 d	2.52 ± 0.12 b	3.60 ± 0.09 bc	77.75 ± 0.15 b
	Roots and storage roots	7.15 ± 0.25 b	2.92 ± 0.25 ab	5.23 ± 0.07 a	88.85 ± 0.46 a
	Rhizomes	5.96 ± 0.04 c	1.61 ± 0.14 c	4.03 ± 0.04 b	52.20 ± 0.51 f
Tissue- cultured plants	Leaves	3.98 ± 0.17 e	1.11 ± 0.03 d	4.10 ± 0.52 b	61.31 ± 0.32 d
	Pseudostems	3.76 ± 0.05 e	1.17 ± 0.03 cd	1.53 ± 0.08 d	54.98 ± 0.53 e
	Roots	4.45 ± 0.15 d	1.36 ± 0.09 cd	0.85 ± 0.08 e	54.62 ± 1.06 e

Values are mean ± SE from three technical replicates. Means with the same letters in each column are not significantly different at *p* ≤ 0.05 according to DMRT.

**Table 6 plants-14-03544-t006:** Quantification of phenolic compounds in different plant parts of *Globba sirirugsae* derived from wild plants and in vitro-cultured plants.

Condition	Explant	Phenolic Acid Contents (μg/g DW)							
		CA	CMA	CNA	FA	GA	SA	VA	Total
Wild plants	Leaves	ND	141.33 ± 0.09 d	329.73 ± 7.48 d	258.75 ± 0.28 c	ND	ND	ND	729.81 ± 2.62
	Pseudostems	ND	ND	926.76 ± 22.65 c	251.77 ± 0.46 d	ND	ND	ND	1178.53 ± 11.55
	Roots and storage roots	ND	141.28 ± 0.07 d	1447.67 ± 21.45 a	293.65 ± 0.63 b	ND	ND	ND	1882.60 ± 12.62
	Rhizomes	4.69 ± 1.62 b	138.90 ± 0.01 e	1152.92 ± 16.50 b	400.81 ± 0.72 a	ND	ND	ND	1697.32 ± 8.03
In vitro-cultured plants	Leaves	ND	149.09 ± 0.08 c	125.24 ± 0.60 f	205.19 ± 0.29 g	347.67 ± 0.48 b	917.61 ± 10.81 b	442.12 ± 1.77 a	2186.92 ± 2.34
	Pseudostems	ND	152.94 ± 0.03 b	174.13 ± 0.86 e	208.91 ± 0.75 f	343.83 ± 0.51 c	ND	230.19 ± 2.79 b	1110.00 ± 1.07
	Roots	331.19 ±1.29 a	157.53 ± 0.03 a	166.95 ± 0.61 e	220.46 ± 0.78 e	382.35 ± 0.43 a	1418.02 ± 2.64 a	ND	2676.50 ± 1.12

Values are mean ± SE from three technical replicates. Means with the same letters in each row are not significantly different at *p* ≤ 0.05 according to DMRT. Abbreviations: CA = caffeic acid; CMA = *p*-coumaric acid; CNA = cinnamic acid; FA = ferulic acid; GA = gallic acid; ND = not detected; SA = syringic acid; VA = vanillic acid.

**Table 7 plants-14-03544-t007:** Quantification of flavonoid compounds in different plant parts of *Globba sirirugsae* derived from wild plants and in vitro-cultured plants.

Condition	Explant	Flavonoid Compound Contents (μg/g DW)				
		Catechin	Kaempferol	Quercetin	Rutin	Total
Wild plants	Leaves	ND	1755.62 ± 26.10 a	434.29 ± 15.21 b	36.27 ± 0.17 bc	2226.18 ± 13.83
	Pseudostems	ND	164.95 ± 2.08 e	409.99 ± 3.13 c	31.59 ± 0.05 e	606.53 ± 1.75
	Roots and storage roots	ND	383.76 ± 12.28 c	372.45 ± 1.15 d	84.76 ± 1.53 a	840.97 ± 4.99
	Rhizomes	ND	152.95 ± 0.42 e	233.87 ± 0.71 f	37.73 ± 0.01 b	424.55 ± 0.38
In vitro- cultured plants	Leaves	ND	262.79 ± 1.33 d	65.35 ± 0.41 g	35.29 ± 0.40 cd	363.43 ± 0.71
	Pseudostems	ND	275.35 ± 1.36 d	327.33 ± 1.69 e	34.12 ± 0.28 d	636.80 ± 1.11
	Roots	ND	552.44 ± 0.29 b	802.55 ± 13.43 a	35.34 ± 0.52 cd	1390.33 ± 4.75

Values are mean ± SE from three technical replicates. Means with the same letters in each row are not significantly different at *p* ≤ 0.05 according to DMRT. Abbreviations: ND = not detected.

**Table 8 plants-14-03544-t008:** Volatile composition of various parts of *Globba sirirugsae* under different growth conditions.

No.	Volatile Compound	RT	RI	MF	MW	Peak Area (%)					
						Growth Conditions					
						Wild Plants			In Vitro-Cultured Plants		
						L	Rh	Rs	L	P	R
	Monoterpene Hydrocarbons										
1.	α-Pinene	8.48	933	C_10_H_16_	136.23	19.11	35.72	32.71	5.19	4.65	19.11
2.	Camphene	8.94	948	C_10_H_16_	136.23	0.15	6.28	-	-	0.35	-
3.	Sabinene	9.73	969	C_10_H_16_	136.23	0.22	-	-	-	-	-
4.	β-Pinene	9.80	976	C_10_H_16_	136.23	22.38	54.71	48.61	9.04	11.50	-
5.	β-Myrcene	10.33	993	C_10_H_16_	136.23	-	-	-	-	0.14	-
6.	D-Limonene	11.45	1030	C_10_H_16_	136.23	-	-	-	-	0.50	-
	Oxygenated monoterpenes										
7.	Dehydro-1,8-cineole	10.27	988	C_10_H_16_O	152.23	0.40	-	-	-	-	-
8.	Eucalyptol	11.50	1032	C_10_H_18_O	154.25	2.11	-	-	1.22	-	-
9.	Bornyl acetate	19.65	1290	C_12_H_20_O_2_	196.29	-	-	-	-	0.08	-
10.	Myrtenyl acetate	20.83	1296	C_12_H_18_O_2_	194.27	-	-	-	-	0.07	-
11.	2-Oxabicyclo [2.2.2]octan-6-ol, 1,3,3-trimethyl-, acetate	21.28	1347	C_12_H_20_O_3_	212.29	-	-	-	-	0.30	-
	Sesquiterpene hydrocarbons										
12.	δ-Elemene	21.13	1341	C_15_H_24_	204.35	-	-	-	0.47	0.28	-
13.	α-Cubebene	21.46	1354	C_15_H_24_	204.35	0.18	-	-	-	0.06	-
14.	Cypera-2,4-diene	21.85	1367	C_15_H_22_	202.34	-	-	-	0.57	2.84	-
15.	Aristolediene	22.18	1378	C_15_H_22_	202.34	-	-	-	0.20	2.61	-
16.	Copaene	22.21	1381	C_15_H_24_	204.35	5.56	-	-			-
17.	β-Elemene	22.48	1389	C_15_H_24_	204.35	-	-	-	8.93	5.09	-
18.	Cyperene	22.88	1405	C_15_H_24_	204.35	-	-	-	3.13	8.80	53.18
19.	isoledene	23.29	1420	C_15_H_24_	204.35	-	-	-	0.20	0.82	-
20.	Caryophyllene	23.42	1426	C_15_H_24_	204.35	0.55	0.85	18.68	22.99	8.29	0.55
21.	Selina-5,11-diene	24.05	1450	C_15_H_24_	204.35	0.71	-	-	0.26	-	-
22.	α-Gurjunene	24.18	1455	C_15_H_24_	204.35	2.45	-	-	-	-	-
23.	γ-Maaliene	24.21	1458	C_15_H_24_	204.35	-	-	-	-	1.13	-
24.	Humulene	24.35	1460	C_15_H_24_	204.35	-	-	-	14.71	8.09	-
25.	Alloaromadendrene	24.52	1468	C_15_H_24_	204.35	2.29	-	-	-	-	-
26.	cis-Muurola-4(15),5-diene	24.59	1470	C_15_H_24_	204.35	-	-	-	-	0.56	-
27.	γ-Gurjunene	24.88	1478	C_15_H_24_	204.35	2.26	-	-	-	-	-
28.	α-Gurjunene	24.89	1481	C_15_H_24_	204.35	-	-	-	2.68	4.43	-
29.	α-Panasinsen	24.96	1484	C_15_H_24_	204.35	-	-	-	-	1.86	-
30.	β-Copaene	25.04	1485	C_15_H_24_	204.35	0.24	-	-	-	-	-
31.	β-Cubebene	25.06	1488	C_15_H_24_	204.35	-	-	-	4.38	3.24	-
32.	Aristolochene	25.13	1491	C_15_H_24_	204.35	2.71	-	-	0.93	1.20	-
33.	β-Selinene	25.18	1493	C_15_H_24_	204.35	4.75	-	-	2.10	2.44	-
34.	α-Bergamotene	25.36	1496	C_15_H_24_	204.35	18.98	-	-	-	-	-
35.	Eremophylene	25.39	1501	C_15_H_24_	204.35	-	-	-	2.48	4.47	-
36.	δ-Guaiene	25.66	1510	C_15_H_24_	204.35	6.05	-	-	-	-	-
37.	γ-Cadinene	25.89	1521	C_15_H_24_	204.35	-	-	-	0.40	1.83	-
38.	α-Selinene	25.99	1525	C_15_H_24_	204.35	2.35	-	-	3.15	5.25	-
39.	β-Cadinene	26.01	1526	C_15_H_24_	204.35	1.08	-	-	0.28	1.23	-
40.	α-Guaiene	26.12	1531	C_15_H_24_	204.35	-	-	-	-	7.09	-
41.	α--Himachalene	26.12	1531	C_15_H_24_	204.35	-	-	-	1.33		-
42.	α-Cadinene	26.48	1158	C_15_H_24_	204.35	-	-	-	-	0.07	-
43.	1,5-Cyclodecadiene, 1,5-dimethyl-8-(1-methylethylidene)-, (E,E)-	26.98	1566	C_15_H_24_	204.35	-	-	-	-	0.12	-
44.	Aromadendrene	28.77	1643	C_15_H_24_	204.35	-	-	-	-	0.26	-
	Oxygenated sesquiterpenes										
45.	10-epi-Elemol	26.78	1558	C_15_H_26_O	222.37	-	-	-	-	0.19	-
46.	1-Naphthalenol, 1,2,3,4,4a,7,8,8a-octahydro-1,6-dimethyl-4-(1-methylethyl)-, [1R-(1α,4β,4aβ,8aβ)]-	27.85	1602	C_15_H_26_O	222.37	-	-	-	-	0.11	-
47.	Epicubenol	28.37	1625	C_15_H_26_O	222.37	-	-	-	-	0.53	-
48.	tau-Cadinol	28.63	1636	C_15_H_26_O	222.37	-	-	-	-	0.09	-
49.	Selina-6-en-4-ol	28.94	1650	C_15_H_26_O	222.37	-	-	-	-	0.30	-
50.	tau-Muurolol	29.24	1663	C_15_H_26_O	222.37	-	-	-	-	0.24	-
51.	Neointermedeol	29.31	1667	C_15_H_26_O	222.37	-	-	-	-	3.27	-
52.	11-Selinene-4-ol	29.41	1671	C_15_H_26_O	222.37	-	-	-	-	4.06	-
53.	Ambrial	32.54	1816	C_16_H_26_O	234.38	-	-	-	-	0.08	-
	Miscellaneous										
54.	Butanal, 2-methyl-	2.59	682	C_5_H_10_O	86.13	3.24	-	-	2.16	0.27	26.16
55.	Acetic acid	2.92	699	C_2_H_4_O_2_	60.05	-	-	-	1.97	0.30	-
56.	Pyrrole	4.16	766	C_4_H_5_N	67.09	-	-	-	-	0.11	-
57.	Octamethyltetrasiloxane	10.51	999	C_8_H_24_O_4_Si_4_	296.62	-	-	-	0.20	-	-
58.	endo-2-Chlorobornane	15.70	1190	C_10_H_17_Cl	172.70		1.18	-	-	-	-
59.	3-Nonen-5-yne, 4-ethyl-	21.10	1341	C_11_H_18_	150.26	2.22	-	-	-	-	-
60.	Tetradecamethylcycloheptasiloxane	25.31	1497	C_14_H_42_O_7_Si_7_	519.08	-	-	-	0.27	-	-
61.	Octadecamethylcyclononasiloxane	32.71	1824	C_18_H_54_O_9_Si_9_	667.39	-	-	-	1.25	-	-
62.	Eicosamethyl-cyclodecasiloxane	35.78	1975	C_20_H_60_O_10_Si_10_	741.54	-	-	-	0.14	-	-
	Monoterpene hydrocarbons [%]					41.86	96.71	81.32	14.23	17.14	19.11
	Oxygenated monoterpenes [%]					2.51	-	-	1.22	0.45	-
	Sesquiterpene hydrocarbons [%]					50.16	0.85	18.68	69.19	72.06	53.73
	Oxygenated sesquiterpenes [%]					-	-	-	8.87	-	-
	Miscellaneous [%]					5.46	1.18	-	5.99	0.68	26.16
	Number of constituents					22	6	3	27	44	4

L = leaves; MF = molecular formula; MW = molecular weight; P = pseudostems; R = roots; RI = retention index; Rh = rhizomes; Rs = roots and storage roots; RT = retention time.

**Table 9 plants-14-03544-t009:** FTIR profile of *Globba sirirugsae* from wild plants and micropropagated plants.

Wavenumber (cm^−1^)							Vibrational Type	Corresponding Functional Groups
Wild Plants				MS (2 mg/L BA + 0.1 mg/L NAA)				
L	P	Rs	Rh	L	P	R		
3321	3322	3330	3326	3320	3332	3321	O–H (Stretching)	Alcohols, Phenols
2929	2919	2920	2919	2918	2919	2920	C–H (Stretching)	–CH_2_ (Methylene) (Cellulose)
2850	2851	2848	2851	2850	2851	2852	C–H (Stretching)	–CH_2_ (Methylene) (Cellulose)
1735	1730	1734	1730	1731	1731	1735	C=O (Stretching)	Ester, Aldehyde
1635	1623	1642	1631	1632	1632	1633	C=C (Stretching)	Aromatic
1521	1518	1516	1516	1524	1519	1517	C=C (Stretching)	Aromatic
-	-	-	-	1426	1426	1425	C–H (bending)	CH_2_, CH_3_
-	-	-	-	1398	1398	1396	C–H (bending)	CH_2_, CH_3_
1370	1365	1365	1368	1361	1360	1360	C–H (bending), O–H (bending)	–CH_3_, Alcohols, Phenols
1313	1311	1320	1320	1314	1314	1315	O–H (bending), C–O (Stretching)	Phenolics, Flavonoids, Cellulose
1244	1236	1236	1244	1236	1235	1236	C–O (Stretching)	Phenols, Cellulose
1033	1032	1032	1032	1034	1032	1032	C–O (Stretching)	Alcohol, Cellulose

L = leaves; MS = micropropagated plants, P = pseudostems; R = roots; Rh = rhizomes; Rs = root and storage roots.

**Table 10 plants-14-03544-t010:** Formulations of PGR components combined with solid MS medium for in vitro plant multiplication of *Globba sirirugsae*.

No.	PGR Formulation	PGRs (mg/L)	
1.	BA combined with NAA (11 treatments)	BA	0, 1, 2, 3, 4, 5
		NAA	0, 0.1
2.	kinetin combined with NAA (11 treatments)	kinetin	0, 1, 2, 3, 4, 5
		NAA	0, 0.1
3.	BA combined with NAA (13 treatments)	BA	0, 0.5, 1, 2, 3, 4, 5
		NAA	0, 0.5, 1
4.	kinetin combined with NAA (13 treatments)	kinetin	0, 0.5, 1, 2, 3, 4, 5
		NAA	0, 0.5, 1

All experimental controls were cultured on PGR-free solid MS medium.

**Table 11 plants-14-03544-t011:** HPLC operating conditions to identify the phenolic acids and flavonoids.

Parameter	Operating Conditions
Mobile phase	Line A: 0.1% acetic acid in dH_2_O Line B: acetonitrile
Detector	UV-diode array; 280 nm for hydroxybenzoic acid, 320 nm for hydroxy cinnamic acid, 370 nm for flavonoids
Column	C–18
Column temperature	38 °C
Flow rate	0.8 mL/min
Sample injection volume	20 µL

## Data Availability

All data are contained within the article.

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
