# Peer review of "In Vitro Regeneration, Acclimatization, Phytochemical Profiling, and Antioxidant Properties of Hong Hoen Sirirugsa (Globba sirirugsae Saensouk & P.Saensouk)"

_plants, 2025, doi:10.3390/plants14223544_

Round 1

Reviewer 1 Report

Comments and Suggestions for Authors

The article presents a comprehensive and systematic study on the in vitro regeneration, acclimatization, and phytochemical profile of Globba sirirugsae. The integrated analysis of tissue culture and phenolic, flavonoid, and volatile compounds, particularly on a rare Zingiberaceae species, makes a unique contribution to the literature. The findings shed light on the plant's pharmaceutical and ornamental potential and provide important data for conservation biotechnology. Overall, the study is well-structured, its methodology is clear, the data are consistent, and the discussion section is consistent with the literature. However, some sections could benefit from improvement in terms of data interpretation, graphical presentation, and clarity of statistical analysis.

My evaluations of the article are provided below.

1- While the introduction provides a comprehensive overview of the literature, the originality of the study and how it fills a scientific gap should be more clearly emphasized.

2- Although the software and parameters used for the ANOVA and DMRT tests are specified in the methodology section, it is not explained whether prerequisites such as normal distribution and variance homogeneity were checked.

3- The tables are quite detailed but very dense; Table 8 (GC–MS results), in particular, is overcrowded, so it may be appropriate to move it to the supplementary material.

4- While the discussion section is sufficient for comparing the results with existing literature, more critical comments could be added regarding biotechnological applications (e.g., industrial scaling of metabolite production, commercial importance of tissue culture).

5- The quality and resolution of the chromatograms or spectra from HPLC, FTIR, and GC–MS analyses should be improved; axis naming and scale information are unclear in some figures.

6- While the English language is generally fluent, repeated expressions in some sections (such as "by contrast" or "similarly") could be simplified. The numbering of the "Materials and Methods" subheadings should be revised according to journal format.

7- The results provide a strong summary; however, clear research directions for future studies (e.g. genetic analysis of metabolite biosynthesis pathways, bioreactor applications, etc.) should be suggested more clearly.

Comments on the Quality of English Language

While the English language is generally fluent, repeated expressions in some sections (such as "by contrast" or "similarly") could be simplified. The numbering of the "Materials and Methods" subheadings should be revised according to journal format.

Author Response

  1. While the introduction provides a comprehensive overview of the literature, the originality of the study and how it fills a scientific gap should be more clearly emphasized.

Respond comment 1: Thank you for your thoughtful suggestion. We have carefully reviewed the manuscript and revised the introduction accordingly. Specifically, we clarified the knowledge gap and emphasized the significance of our study by stating:

Although G. sirirugsae is horticulturally valuable and potentially useful as a medicinal herb, its in vitro propagation and phytochemical profile remain unexplored. This gap limits both conservation efforts and commercial development. To address this, we present the first protocol for in vitro propagation and phytochemical profiling of this endemic species, supporting its sustainable utilization and long-term preservation. (line 78-82).

We hope this revision clearly highlights the rationale and relevance of our research.

  1. Although the software and parameters used for the ANOVA and DMRT tests are specified in the methodology section, it is not explained whether prerequisites such as normal distribution and variance homogeneity were checked.

Respond comment 2: Thank you for your observation. Datasets were analyzed for normality by Shapiro–Wilk test and homogeneity by Levene’s test followed by Analysis of Variance (ANOVA) with the following degree of freedom. We have now included the statistical analysis section as recommended (line 712-713).

  1. The tables are quite detailed but very dense; Table 8 (GC–MS results), in particular, is overcrowded, so it may be appropriate to move it to the supplementary material.

Respond comment 3: Thank you for your thoughtful suggestion. While we appreciate the concern regarding table density, we believe retaining Table 8 in the main manuscript is appropriate, as it presents essential compound data by chemical class—such as monoterpenes, sesquiterpenes, and phenolic derivatives—which directly support our interpretation. Relocating it to the supplementary material may disrupt the flow and hinder readers seeking compound-specific insights during discussion.

  1. While the discussion section is sufficient for comparing the results with existing literature, more critical comments could be added regarding biotechnological applications (e.g., industrial scaling of metabolite production, commercial importance of tissue culture).

Respond comment 4: Thank you for the valuable suggestion. We have revised the discussion section to include additional comments on the biotechnological applications of our findings, particularly regarding industrial-scale metabolite production and the commercial relevance of tissue culture systems (line 209-213, 258-264, 402-410).

  1. The quality and resolution of the chromatograms or spectra from HPLC, FTIR, and GC–MS analyses should be improved; axis naming and scale information are unclear in some figures.

Respond comment 5: We appreciate your suggestion about the quality and clarity of the chromatograms and spectra. In response, we have revised the figures from HPLC, FTIR, and GC–MS analyses to enhance resolution and improve overall readability. Axis labels and scale indicators have also been clarified to ensure consistency and interpretability.

  1. While the English language is generally fluent, repeated expressions in some sections (such as "by contrast" or "similarly") could be simplified. The numbering of the "Materials and Methods" subheadings should be revised according to journal format.

Respond comment 6: Thank you for your suggestion regarding language clarity and formatting. We have reviewed the manuscript and revised repetitive expressions to improve readability. Additionally, the numbering of the materials and methods subheadings has been adjusted to comply with the journal’s formatting guidelines.

  1. The results provide a strong summary; however, clear research directions for future studies (e.g. genetic analysis of metabolite biosynthesis pathways, bioreactor applications, etc.) should be suggested more clearly.

Respond comment 7: Thank you for the helpful suggestion. We have revised the discussion section to clarify future research directions, including genetic analysis of metabolite biosynthesis pathways and potential bioreactor applications for scalable production (line 209-213, 258-264, 402-410).

Comments on the Quality of English Language

While the English language is generally fluent, repeated expressions in some sections (such as "by contrast" or "similarly") could be simplified. The numbering of the "Materials and Methods" subheadings should be revised according to journal format.

Respond comment: We acknowledge the reviewer’s observations regarding recurring transitional phrases and formatting inconsistencies. To address these points, we have refined the language to reduce redundancy and enhance flow. The subheading structure within the materials and methods section has also been revised to align with the journal’s formatting standards.

Reviewer 2 Report

Comments and Suggestions for Authors

The current paper described plant regeneration from Globba sirirugsae.

The results also include analysis of secondary metabolites, phenolic compounds and antioxidants capacity.

Authors performed a good work, however discussion still based on old concept of exogenous auxin /cytokinin ratio.

The text is well-written, but discussion require significant corrections.

Details:

Lines 29, 78: please, provide in the text evidences that Globba require N:P:Cl ratio 48:1:5 – one you used in your paper.

Line 35: “DPPH and ABTS” – please, avoid such abbreviations in summary without explanations.

Fig.1 – it will great add scale bar for each image.

Line 63: “plant growth regulators (PGRs) or hormones” ?? Exogenous? Why “or”? Please, remember that plant morphogenesis regulated by endogenous one only.

Line 103: “Cytokinins play a fundamental role in plant development” – you used citation from 2001, while currently more knowledges come.

Line 111: “hormonal balance between cytokinins and auxins” – do you mean exogenous or endogenous?  It seems you describe old, not valid anymore of auxin/cytokinin ratio. It will be great to re-formulate by correct way (https://doi.org/10.3390/ijpb16030097). I would  suggest correct one: “Excess exogenous BAP lead to excess auxin production and disrupt balance between auxin production and canalization/polar auxin transport”.

Line 122: “Moderate cytokinin or BA concentrations” - ??  Exogenous cytokinin. BA is cytokinin analogue.

Line 132: “plant induction” – please, reformulate.  

Line 134: “by kinetin” = by exogenous kinetin.

Line 186: “shoot development was influenced by the cytokinin-to-auxin balance”? please, use correct statement instead old concept of auxin/cytokinin balance.

Line 190: “Skoog and Miller [28] found that explants treated with an increased auxin-to-cytokinin ratio promoted root development, whereas a decreased ratio enhanced shoot formation.” ?? Please, consider that both shoot and root were regulated only by endogenous auxin and it canalization.

Line 263: “minimal environmental stress” – in vitro you probably used high nutritional stress.

Table 5: very good idea.

Line 477: “were transferred onto Murashige and Skoog (1962) (MS) médium” ? please, clarify light and temperature.

Author Response

Comment 1. Lines 29, 78: please, provide in the text evidences that Globba require N:P:Cl ratio 48:1:5 – one you used in your paper.

Response 1: Thank you for your comment. The N:P:Cl ratio of 48:1:5 reflects the macronutrient composition of the MS medium used in our study. Although this N:P:Cl ratio was not specifically optimized for Globba sirirugsae, it has been widely applied in micropropagation protocols for Globba species and other Zingiberaceae species.

Comment 2.  Line 35: “DPPH and ABTS” – please, avoid such abbreviations in summary without explanations.

Response 2: Thank you for your suggestion. To enhance clarity, we have revised the abstract by spelling out the full names of DPPH and ABTS at their first mention (lines 35–36).

Comment 3.  Fig.1 – it will great add scale bar for each image.

Response 3: Thank you for your suggestion. Scale bars have been added to Figure 1, with clarification included in the legend (line 62-63).

Comment 4.  Line 63: “plant growth regulators (PGRs) or hormones” ?? Exogenous? Why “or”? Please, remember that plant morphogenesis regulated by endogenous one only.

Response 4: Thank you for your observation. We agree that the distinction between exogenous plant growth regulators (PGRs) and endogenous hormones should be made clearer.
The sentence has been revised to specify that kinetin and NAA were applied as exogenous PGRs to influence endogenous hormonal pathways involved in morphogenesis.

The word “or” has been removed to avoid confusion The revised sentence is exogenous plant growth regulators (PGRs) optimize growth and multiplication (line 65-66).

Comment 5: Line 103: “Cytokinins play a fundamental role in plant development” – you used citation from 2001, while currently more knowledges come.

Response 5: Thank you for your valuable comment. We have revised the manuscript to clarify the role of cytokinins are key regulators in plant tissue culture, promoting cell division, shoot regeneration, and developmental plasticity. Their application enhances in vitro propagation by activating a network of shoot-inducing genes and restoring totipotency in cultured explants (line 120-123).

Comment 6: Line 111: “hormonal balance between cytokinins and auxins” – do you mean exogenous or endogenous?  It seems you describe old, not valid anymore of auxin/cytokinin ratio. It will be great to re-formulate by correct way (https://doi.org/10.3390/ijpb16030097).
I would  suggest correct one: “Excess exogenous BAP lead to excess auxin production and disrupt balance between auxin production and canalization/polar auxin transport”.

Response 6: Thank you for your valuable suggestion. We agree that the classical auxin/cytokinin ratio model. Accordingly, the sentence has been revised to:

In contrast, excessive exogenous BA leads to overproduction of auxin, disrupting the balance between auxin biosynthesis and canalization/polar auxin transport. (100-105).

Comment 7: Line 122: “Moderate cytokinin or BA concentrations” - ??  Exogenous cytokinin. BA is cytokinin analogue.

Response 7: Thank you for pointing this out. We agree that BA is a synthetic cytokinin analogue and should be referred to as exogenous cytokinin. The sentence has been revised:

Moderate exogenous BA concentrations (line 118).

Comment 8: Line 132: “plant induction” – please, reformulate.  

Response 8: Thank you for your suggestion. We agree that the term plant induction was too general. The figure caption has been revised to specify the developmental process observed.
The updated caption is: Figure 2. Effect of BA and NAA on in vitro plantlet regeneration of Globba sirirugsae. Scale bar = 2 cm. Figures 3 has also been revised accordingly (line 136 and line 178).

Comment 9: Line 134: “by kinetin” = by exogenous kinetin.

Response 9: Thank you for the comment. We agree and have revised the phrase to “by exogenous kinetin” for clarity (line 139).

 Comment 10: Line 186: “shoot development was influenced by the cytokinin-to-auxin balance”? please, use correct statement instead old concept of auxin/cytokinin balance.

Response 10: Thank you for your suggestion. We have revised the statement to suggest that the superior efficacy of exogenous BA over kinetin in shoot development is attributable to its stronger cytokinin signaling, which activates key shoot-promoting genes (line 196-198).

Comment 11: Line 190: “Skoog and Miller [28] found that explants treated with an increased auxin-to-cytokinin ratio promoted root development, whereas a decreased ratio enhanced shoot formation.” ?? Please, consider that both shoot and root were regulated only by endogenous auxin and it canalization.

Response 11: Thank you for your insightful comment. We acknowledge that the classical model proposed by Skoog and Miller emphasized the role of the exogenous auxin-to-cytokinin ratio in organogenesis. However, we agree that current understanding also highlights the importance of endogenous auxin gradients and canalization in regulating shoot and root formation. This point has been revised accordingly in lines 202–208.

Comment 12: Line 263: “minimal environmental stress” – in vitro you probably used high nutritional stress.

Response 12: We thank the reviewer for the clarification. The term “minimal environmental stress” has been revised to better reflect the in vitro context. We described tissue culture conditions provide a highly controlled with defined abiotic inputs including artificial nutrient and exogenous hormone regimes that may impose nutritional stress distinct from natural habitat (line 290-292 and line 322-325).

Comment 13: Table 5: very good idea.

Response 13: Thank you for your positive feedback. We are glad that Table 5 was helpful in presenting the data clearly.

Comment 14: Line 477: “were transferred onto Murashige and Skoog (1962) (MS) médium” ? please, clarify light and temperature.

Response 14: Thank you for the valuable suggestion. The light and temperature conditions have now been added to the revised sentence (lines 518–519).

Reviewer 3 Report

Comments and Suggestions for Authors

Several methodological and interpretative issues prevent a clear assessment of your findings and can be corrected with targeted revisions at specific places in the manuscript. In Section 2.12 Statistical Analysis, specify for each experiment whether a one-way or two-way ANOVA was used, state the exact model structure and error term, and indicate how assumptions of normality and homogeneity of variances were checked before applying DMRT. Clarify explicitly whether the reported replicates in the regeneration experiments and in the biochemical assays are biological or technical, and reflect this consistently in the captions of all tables and figures. In Section 2.10 GC–MS Analysis, add the retention index information for the principal compounds and state the minimum NIST library match threshold accepted for identification. Confirm whether peak areas were normalized and reported as relative percentages. Without these details, the reliability of the compound assignments cannot be evaluated. In Section 2.11 FTIR Analysis, include a brief note on spectral pre-processing, indicating whether baseline correction, smoothing, or normalization was applied prior to interpretation. Harmonize the reporting of antioxidant results. Tables and text currently mix milligrams of Trolox equivalents per gram dry weight with percentage inhibition. Choose a single unit system for all assays and apply it throughout, including in table headings and figure captions, so that results are directly comparable. Strengthen interpretation in the Results and Discussion where the text following Tables 1 to 3 largely restates numbers. Explain why specific BA/NAA or KN/NAA combinations produced superior regeneration in the context of cytokinin–auxin interactions, and link observed differences in phenolic and flavonoid contents to plausible biosynthetic responses to in vitro culture conditions. Where numerical values in the text differ slightly from those in Table 5 for total phenolics and flavonoids, correct the discrepancies so that all sections report identical values. Improve figure quality and captions. Replace Figures 2 to 5 with higher-resolution images, add unified scale bars, and state the number of replicates per treatment directly in the captions together with a clear explanation of the lettering used for DMRT groupings. For Figure 6, report the sample size used to compute Pearson’s correlations so that readers can judge the robustness of the relationships. Streamline presentation of the chemical profiles. Keep only the major HPLC and GC–MS findings in the main text using concise summary graphics, and move full compound lists to the Supplementary Material. In the discussion of cinnamic acid, ferulic acid, kaempferol, and quercetin, connect these metabolites directly to the antioxidant activities measured in your samples rather than relying on general pharmacological descriptions from the literature. Finally, standardize unit notation across the manuscript, ensure consistent decimal formatting in all tables, and edit the prose for concision and precision by reducing repetitive sentence openers and long multi-clause sentences. These concrete changes, made at the indicated sections, will substantially improve transparency, interpretability, and overall scientific rigor.

Author Response

Section 2.12 Statistical Analysis

  • Specify for each experiment whether a one-way or two-way ANOVA was used.

Responds reviewer: We thank the reviewer for the suggestion. One-way ANOVA was applied to both experimental series (in vitro propagation and phytochemical content analyses) because each analysis compared mean values across multiple levels of a single factor (PGRs formula or independent sample groups). Each sample set was treated as independent for the purposes of statistical analysis. Duncan’s Multiple Range Test (DMRT) was used as a post hoc test to identify pairwise differences among group means with significance at p ≤ 0.05.

  • State the exact model structure and error term.

Responds reviewer: Thank you for your comment. The model used for the one-way ANOVA was defined as:

where Yij  is the observed value of the dependent variable for the j observation in the i group, μ is the overall mean, αi is the fixed effect of the i treatment (e.g., PGRs formula), and εij is the residual error term assumed to be independently and normally distributed with mean zero and constant variance. The error term used for the F-test was the within-group mean square.

  • Indicate how assumptions of normality and homogeneity of variances were checked before applying DMRT.

Responds reviewer: Thank you for your comment. Prior to applying DMRT, assumptions of normality and homogeneity of variances were evaluated (via SPSS software). Normality of residuals was assessed using the Shapiro–Wilk test (p ≥ 0.05). For Homogeneity of variances was tested using Levene’s test (p ≥ 0.05). Variables that met both assumptions (p ≥ 0.05) were analyzed using one-way ANOVA (sig. < 0.05) followed by DMRT.

  • Clarify explicitly whether the reported replicates in the regeneration experiments and in the biochemical assays are biological or technical, Reflect this consistently in the captions of all tables and figures.

Responds reviewer: Thank you for your observation and suggestions. We agree with those suggestions and have revised all tables and figures accordingly.

 Section 2.10 GC–MS Analysis

  • Add the retention index information for the principal compounds.

Responds reviewer: Thank you for your suggestion, retention index (RI) information has now been added for the principal compounds (Table 8).

  • State the minimum NIST library match threshold accepted for identification.

Responds reviewer: In this study, compounds were identified based on spectral matching with the NIST library. A minimum match threshold of 800.0 was accepted for compound identification.

  • Confirm whether peak areas were normalized.
    Responds reviewer: We confirm data were normalized.

  • Confirm whether peak areas were reported as relative percentages.

Responds reviewer: We confirm reported as relative percentages.

Section 2.11 FTIR Analysis

  • Include a brief note on spectral pre-processing, indicating whether baseline correction, smoothing, or normalization was applied prior to interpretation.

Responds reviewer: Thank you for your suggestion. Spectral pre-processing (e.g., baseline correction, smoothing, or normalization) was not applied, as the raw spectra exhibited clear and well-defined peaks that allowed direct interpretation. Should further processing be required, we are prepared to apply appropriate methods and revise the analysis accordingly.

Antioxidant Results Reporting

  • Harmonize the reporting of antioxidant results, choose a single unit system for all assays, apply it throughout, including in table headings and figure captions.

Responds reviewer: Thank you for your valuable suggestion. We have revised the manuscript to harmonize the reporting of antioxidant activity results by adopting a single unit system across all assays. Specifically, antioxidant activity is now consistently expressed as mg Trolox equivalent/gram of extract (mg TE/g) for the DPPH assay and percentage of inhibition for the ABTS assay.

Results and Discussion

  • Strengthen interpretation in the Results and Discussion where the text following Tables 1 to 3 largely restates numbers.

Response to comment: Thank you for your constructive feedback regarding the interpretation of results about In vitro propagation sections. We have revised the corresponding sections in the results and discussion to go beyond numerical restatement by incorporating mechanistic insights, comparative analysis, and references to previous studies. Specifically, we now discuss the hormonal interactions underlying shoot and root regeneration, highlight the concentration-dependent effects of BA and kinetin, and relate our findings to regeneration patterns observed in other Globba species.

  • Explain why specific BA/NAA or KN/NAA combinations produced superior regeneration in the context of cytokinin–auxin interactions.

Response to comment: Thank you for your valuable suggestion. We have revised the discussion section to clarify why specific BA/NAA and KN/NAA combinations produced superior shoot regeneration. As noted in the revised text, the enhanced efficacy of BA over kinetin may be attributed to its stronger cytokinin signaling capacity, which more effectively activates shoot-promoting gene networks. BA has been reported to induce higher expression of cytokinin-responsive genes involved in meristem activation and organogenesis, whereas kinetin may exhibit comparatively weaker signaling strength. This mechanistic distinction helps explain the superior shoot induction observed in BA-treated explants (line 196-198).

  • Link observed differences in phenolic and flavonoid contents to plausible biosynthetic responses to in vitro culture conditions.

Response to comment: Thank you for your valuable comment. We have revised the Discussion section to address the observed differences in phenolic and flavonoid contents in relation to plausible biosynthetic responses under in vitro culture conditions.

The sentence has been revised to: By contrast, tissue culture conditions provide a highly controlled environment with defined abiotic inputs including artificial nutrients and exogenous hormone regimes that may impose nutritional stress distinct from natural habitat. These abiotic factors can trigger oxidative stress and subsequently upregulate key biosynthetic enzymes such as phenylalanine ammonia-lyase (PAL) and chalcone synthase (CHS), leading to enhanced accumulation of antioxidant compounds including phenolics and flavonoids (line 293-296).

  • Correct discrepancies where numerical values in the text differ slightly from those in Table 5 for total phenolics and flavonoids.

Response to comment: Thank you for your observation. We have carefully reviewed the manuscript and corrected all discrepancies between the numerical values reported in the text and those presented in Table 5 for total phenolic and flavonoid contents. The revised text now accurately reflects the data shown in the table, ensuring consistency and clarity throughout the results and discussion sections.

Figures and Captions

  • Improve figure quality and captions, replace Figures 2 to 5 with higher-resolution images add unified scale bars, state the number of replicates per treatment directly in the captions, provide a clear explanation of the lettering used for DMRT groupings, for Figure 6, report the sample size used to compute Pearson’s correlations.

Responds reviewer: Thank you for the valuable suggestions. All figures (Figures 2 to 5) have been replaced with high-resolution versions (600 dpi) and now include unified scale bars. Figure captions have also been revised to provide more detailed descriptions. In addition, the sample size used for Pearson’s correlation analysis has been added directly to the caption of Figure 6 with revised results and discussion (line 335-339, 347-348).

Chemical Profile Presentation

  • Streamline presentation of the chemical profiles, keep only the major HPLC and GC–MS findings in the main text using concise summary graphics, move full compound lists to the supplementary material.

Responds reviewer: Thank you for the suggestion regarding streamlining the chemical profile presentation. While we agree that concise summary graphics improve clarity, we believe that retaining the full compound tables in the main manuscript is important for interpretability. The detected compounds are directly linked to the biological activities discussed and separating them into supplementary material may obscure the connection between metabolite identity and functional relevance. Additionally, fragmenting the data across multiple files could lead to confusion when interpreting the integrated profiles.

Metabolite Discussion

  • In the discussion of cinnamic acid, ferulic acid, kaempferol, and quercetin, connect these metabolites directly to the antioxidant activities measured in your samples, avoid relying on general pharmacological descriptions from the literature.

Responds reviewer: Thank you for your valuable suggestion. We have revised the discussion to directly link cinnamic acid, ferulic acid, kaempferol, and quercetin to the antioxidant activities measured in the sample (line 387-393)

Manuscript Formatting and Style

  • Standardize unit notation across the manuscript, ensure consistent decimal formatting in all tables, edit the prose for concision and precision by reducing repetitive sentence openers and long multi-clause sentences.

Responds reviewer: Thank you for the constructive feedback. We have carefully revised the manuscript to standardize unit notation throughout, ensuring consistency in SI formatting and spacing. All tables have been updated to present decimal values with uniform precision. Additionally, we have edited the prose for concision and clarity by removing repetitive sentence openers and simplifying long multi-clause constructions. These changes improve readability and alignment with journal standards.

Round 2

Reviewer 2 Report

Comments and Suggestions for Authors

Thank you very much for corrections and clarification.

Line 434: “in vitro-cultured plants exhibited a broader volatile” – very good points. Please, consider that secondary metabolites can be regulated by nutritional balance. Changes in nutritional balance led to significant changes in metabolites.

There are only few minor points need further clarity.

Despite some people used so-called MS medium, this medium induced high nutritional stress and suitable rather for soft vacuolated cell formation (hyperhydricity). Please, next time use normal medium.    

Line 519: I think light intensity is too low. Standard in PTC is 80-120. Please, clarify

Author Response

Thank you very much for taking the time to review this manuscript. Please find the detailed

responses below and the corresponding revisions/corrections in the re-submitted files. 

Please see the responses in the attachment.

Reviewer 3 Report

Comments and Suggestions for Authors

Previous comments have been satisfactorily addressed.

Author Response

We appreciate the reviewer’s constructive feedback, which has helped us improve the clarity and quality of the manuscript.